# SOCIAL AGENTS: COLLECTIVE INTELLIGENCE IMPROVES LLM PREDICTIONS

**Aanisha Bhattacharyya**[*] 🔴 🔵 🟢   **Abhilekh Borah**[*] 🔴   **Yaman Kumar Singla**[*] 🔴

**Rajiv Ratn Shah** 🟢   **Changyou Chen** 🔵   **Balaji Krishnamurthy** 🔴

🔴 **Adobe Media and Data Science Research (MDSR)**
🟢 **IIIT-Delhi,** 🔵 **SUNY at Buffalo**
✉ **behavior-in-the-wild@googlegroups.com**

## ABSTRACT

In human society, collective decision making has often outperformed the judgment of individuals. Classic examples range from estimating livestock weights to predicting elections and financial markets, where averaging many independent guesses often yields results more accurate than those of experts. These successes arise because groups bring together diverse perspectives, independent voices, and distributed knowledge, so that idiosyncratic errors average out across independent estimates rather than compound. This principle, known as the Wisdom of Crowds, underpins forecasting in domains from finance to politics. Large Language Models (LLMs), however, typically produce a single definitive answer. While effective in many settings, this uniformity overlooks the diversity of human judgments that shapes how people respond to ads, videos, and webpages. Inspired by how societies benefit from diverse opinions, we ask whether LLM predictions can be improved by simulating many diverse answers rather than one. We introduce Social Agents, a multi-agent framework that instantiates a synthetic society of human-like personas with diverse demographic (e.g., age, gender) and psychographic (e.g., values, interests) attributes. Each persona independently appraises a stimulus such as an advertisement, video, or webpage, offering both a quantitative score (e.g., click-through likelihood, recall score, likability) and a qualitative rationale. The set of persona opinions mirrors a real human crowd, and aggregating them yields a single estimate closer to the crowd mean. Across eleven behavioral prediction tasks, Social Agents outperforms single-LLM baselines by up to 164% on simple judgments (e.g., webpage likability) and up to 24% on complex interpretive reasoning (e.g., video memorability), both with GPT-4o as the backbone. Averaged across models, gains reach 30.5% on low-level and 9.9% on high-level tasks. The individual persona predictions generated by Social Agents also strongly align with human judgments, reaching Pearson correlations up to 0.71. These results position computational crowd simulation as a scalable, interpretable tool for improving behavioral prediction and supporting behavioral and marketing decisions [§].

## 1  INTRODUCTION

*"The many, of whom none is a good man, may yet, when joined together, be better than those few."*
- Aristotle

The idea that aggregated opinions can surpass the accuracy of a single expert has fascinated thinkers since antiquity. Aristotle captured this insight in Politics (350 BCE) (Aristotle, 2009), observing that ordinary people, taken together, can judge music, poetry, and politics better than trained critics, since each notices a different aspect. He emphasized that diversity of skills, experiences, and insights

---

[*]Equal Contribution.
[§]https://behavior-in-the-wild.github.io/social-agents

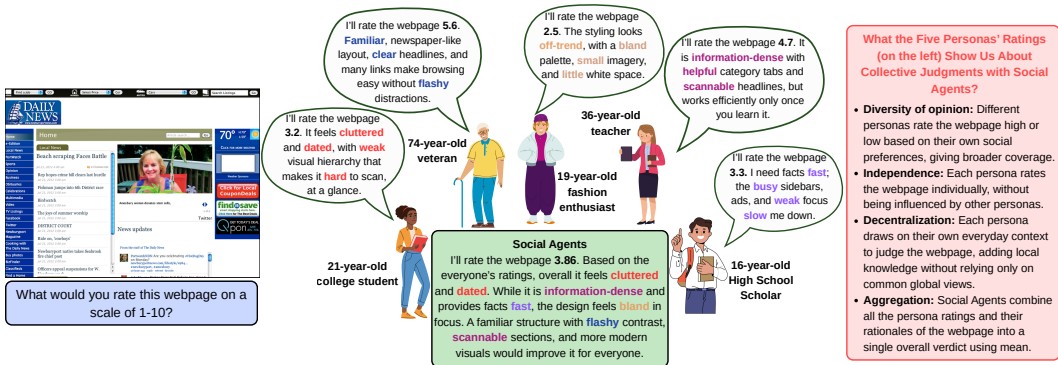

Figure 1: Social Agents are grounded in the principle of the *Wisdom of Crowds*, the idea that collective judgments can outperform individual ones when based on four key ingredients: (i) Diversity of opinion, (ii) Independence, (iii) Decentralization, and (iv) Aggregation (Surowiecki, 2004). As shown in the figure, to capture diversity of social perspectives in webpage likability judgments, we elicit evaluations from five personas varied across age (to reflect generational viewpoints), profession (to represent occupational contexts), and social preferences (to capture differences in values and tastes). Each persona offers independent ratings and rationales, ensuring varied perspectives, which are then aggregated into one collective verdict. The five outer speech bubbles correspond to these independent persona evaluations, while the green box represents the Social Agents aggregated collective verdict.

enhances the quality of collective judgment and decision-making. This early recognition laid the groundwork for what modern scholars now call the wisdom of crowds (Schwartzberg, 2016).

A classic empirical demonstration came from Galton's 1906 analysis of a livestock fair, where nearly eight hundred participants estimated the weight of an ox. Although individual guesses varied widely, the median was within one percent of the true weight, outperforming even the best single estimate and showing robustness to outliers (Galton, 1907). Building on this, Surowiecki (2004) popularized the idea that group decision-making can outperform individuals when four conditions hold (Figure 1): diversity of perspectives, independence of judgment, decentralization of knowledge, and proper aggregation. Over the past two decades, these principles have been applied in practical systems. One example is Best Buy's 2005 internal prediction market, Tag Trade, where employees used virtual currency to bet on outcomes such as holiday gift-card sales. The market's forecast proved accurate, missing actual sales by just 0.6% (Haas et al., 2016). Modern platforms like Polymarket apply the same principles, showing how collective estimation can surpass experts.

Historically, the wisdom of crowds has yielded remarkable accuracy across domains ranging from livestock competitions to geopolitical forecasting. Yet these successes have relied on human participants, which impose limitations: assembling and incentivizing large groups requires sustained effort, is costly to scale, and cannot easily be applied to every decision-making scenario.

As large language models (LLMs) grow more capable, recent systems such as GPT-4o (Hurst et al., 2024) and DeepSeek-R1 (DeepSeek-AI, 2025) now perform complex reasoning tasks once reserved for humans, excelling in mathematics, code generation, and cross-modal analysis. Many are pre-trained on user forum data such as Reddit and other discussion platforms (Radford et al., 2019; Gokaslan & Cohen, 2019; Demszky et al., 2020). Because such corpora contain first-person posts written by users of widely varying ages, professions, and ideologies, the model is repeatedly exposed to how different demographic and psychographic groups phrase opinions, weigh trade-offs, and react to the same stimuli. This exposure gives LLMs an implicit grasp of how personas behave and how individuals with specific psychographic interests interact. For example, Santurkar et al. (2023) demonstrated that persona prompting can steer LLMs to respond in the voice of a given persona across social and opinion-related questions. Tess et al. (2025) extend this finding to applied research settings. They show that persona prompting allows LLMs to serve as proxy users in psychological and social studies. Together, these studies suggest that when steered to adopt a particular persona, LLMs are capable of embodying that perspective and generating responses consistent with it.

While the wisdom of crowds reliably produces accurate judgments with human groups, scaling it in practice remains costly and logistically difficult. LLMs, however, can be steered to act as

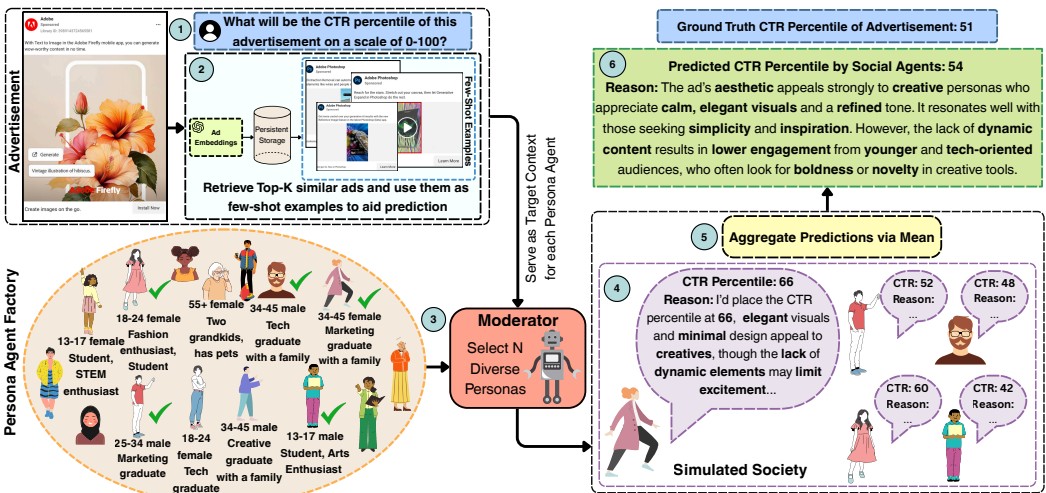

Figure 2: **Overview of the *Social Agents* workflow for Ad Click-Through Rate (CTR) Prediction.** Given an advertisement (top-left), our framework computes its embeddings and retrieves the top-$K$ semantically similar ads from a repository of ad embeddings. These serve as few-shot examples that aid CTR prediction. A *Persona Agent Factory* (bottom-left) contains personas defined by demographic attributes (e.g., age, gender) and traits (e.g., interests, occupation), following templates in Appendix Table 2. From this pool, the *moderator* selects a diverse panel of $N$ personas and instantiates separate LLM agents for each. Each persona agent outputs a CTR percentile (0-100) with a brief rationale. The right-hand side shows the moderator aggregating these predictions via mean to produce a single CTR percentile, with a collective rationale that is compared against the ground-truth CTR percentile.

human-like participants through persona prompting, providing a scalable way to operationalize the wisdom of crowds in synthetic settings. This motivates our central question: *"Can the wisdom of crowds be operationalized through LLMs, where each instance embodies a distinct persona and their aggregated responses improve the predictive and reasoning performance of language models?"*

**Social Agents.** We introduce Social Agents, a multi-agent framework that instantiates a synthetic society of human-like personas with diverse demographic (e.g., age, gender) and psychographic (e.g., values, interests) attributes, operationalizing the Wisdom of Crowds principle through ensembles of LLM-based agents. These agents are deliberately diverse in demographic and psychographic characteristics, ensuring a wide range of perspectives. To preserve independence and avoid groupthink, they are prompted separately. Each agent grounds its reasoning in its assigned profile, providing a decentralized basis for judgment. Their outputs are then aggregated into a collective outcome. In this way, Social Agents capture the four classical pillars of the wisdom of crowds: diversity, independence, decentralization, and aggregation, demonstrating that collective intelligence can improve LLM predictions. This aggregation also offers three advantages. First, *error cancellation* occurs when individual errors offset, bringing the aggregate closer to ground truth. Second, *robustness through diversity* arises as heterogeneous perspectives guard against systematic bias and outliers. Third, *interpretable group dynamics* emerge from the distribution of rationales, revealing drivers of consensus and disagreement. Each agent functions as a simulated evaluator with a unique profile, providing quantitative predictions and qualitative justifications. Aggregating these via a simple mean mitigates variance and bias, improving the accuracy and reliability of population-level predictions.

We conduct a comprehensive empirical evaluation of Social Agents across a suite of challenging tasks. To ground this evaluation, we draw on Construal Level Theory (CLT) (Trope & Liberman, 2010), which posits that psychological distance across temporal, spatial, social, and hypothetical dimensions systematically affects how people represent events. CLT suggests that judgments vary in abstraction: near tasks involve immediate, surface-level evaluations, while distant tasks require deeper semantic elaboration and abstract reasoning. For example, it is straightforward for a user to judge the likability of a webpage based on perceptual cues such as color or layout (a low-level construal), whereas predicting whether the same webpage will be remembered weeks later demands high-level construal, engaging memory, causality, and persuasion. Following this framework, we select 11 diverse tasks spanning the CLT spectrum, including low-level predictions such as click-through rate (CTR), tweet engagement, and behavioral attribute classification (surface: topic, ac-

tion); a medium-level prediction, return on ad spend (ROAS); and high-level constructs such as long-term memorability and behavioral attribute classification (latent: reason, persuasion, emotion). Each task and its dataset are formally defined in Section 2.2.

Following the wisdom of crowds principle, where collective judgments outperform individuals and often experts, our primary comparison is between Social Agents and a *No-Persona* variant, in which a single LLM is prompted as a domain expert. Both variants share the same prompting setup: *5-shot*, where five labeled exemplars are retrieved from the training pool and shown to the model, on every task except Surface and Latent Attribute Classification, which we run *zero-shot* (no exemplars) to remain comparable to prior work on those benchmarks. We further compare against state-of-the-art task-specific expert models, each trained on large labeled corpora: LCBM (Khandelwal et al., 2024), trained on 40,000 YouTube videos and 168 million Twitter posts, is the strongest reported model for CTR, tweet engagement, and tweet content generation; Henry (SI et al., 2023) is the strongest reported model for long-term video memorability; and Behavior-LLaVA (Singh et al., 2025), trained on 730k instruction-response pairs, is the strongest reported model for behavioral attribute classification. For ROAS and webpage likability prediction, two tasks whose targets are continuous numerical scores, we additionally include a tuned XGBoost (Chen & Guestrin, 2016) model, since gradient-boosted trees remain a strong classical baseline on continuous, low-dimensional regression problems. Together, these baselines let us ask whether collective intelligence can improve LLM predictions, offering a scalable alternative to methods reliant on extensive task-specific datasets and training regimes.

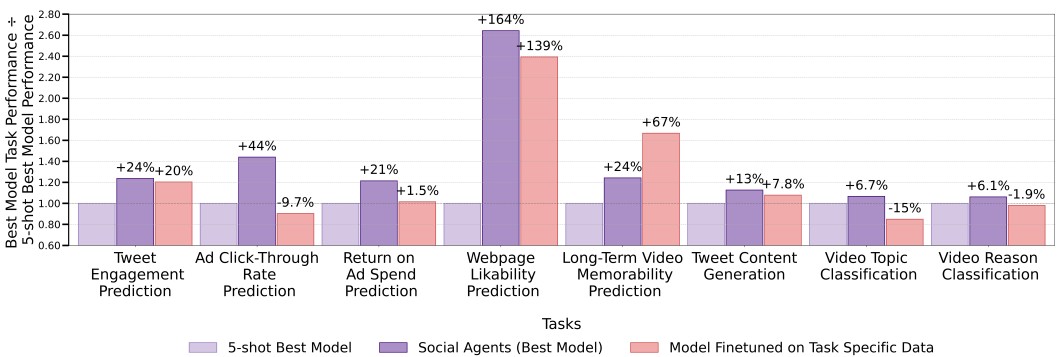

Figure 3: **Performance Comparison of Social Agents (Best Model) over 5-shot Best Model and Models Finetuned on Task-Specific Data across eight tasks.** Across eight tasks, Social Agents (Best Model) consistently improve over the 5-shot Best Model and often exceed models fine-tuned on task-specific data, despite not being trained for those tasks. Here, Best Model refers to whichever base model achieves the strongest results, whether used within Social Agents or in the 5-shot baseline. Performance of Social Agents (Best Model) and fine-tuned baselines is reported relative to a 5-shot Best Model reference (fixed at 1.00). For Models Finetuned on Task Specific Data, we use: Large Content Behavior Models (LCBM) (Khandelwal et al., 2024) for Tweet Engagement Prediction, Ad Click-Through Rate Prediction, and Tweet Content Generation; a trained XGBoost model for Return on Ad Spend Prediction; Henry (SI et al., 2023) for Long-Term Video Memorability Prediction; and Behavior-LLaVA (Singh et al., 2025) for Video Topic and Video Reason Classification tasks.

Empirical results demonstrate that Social Agents improves performance across all construal levels. On low-level tasks involving immediate and behavior-oriented predictions, we observe an average improvement of 30.5% across models, with the largest single-model gain on webpage likability prediction, where GPT-4o performance rises by 164.2% relative to the No-Persona baseline. For the medium-level task of return on ad spend prediction, which measures future financial outcomes such as whether ads will generate revenue, Social Agents achieves a 27.9% average MAPE reduction across both industry domains. On high-level tasks requiring abstract reasoning, the framework yields an average gain of 9.9% across models and tasks. The largest individual gain is on long-term memorability prediction (forecasting whether users will recall an ad later), where GPT-4o improves by 24.2% over the No-Persona baseline. As shown in Fig. 3, Social Agents not only outperforms the No-Persona baseline but also surpasses task-specific expert models in most cases. For GPT-4o specifically, Social Agents reduces MAPE by **34.7%** on Ad CTR (Creative industry) and by **39.8%** on ROAS (Real Estate industry), and raises Pearson $r$ on webpage likability by **164.2%** over the No-Persona baseline. Averaged across model backbones and industries, the gains are **21.75%** on

tweet engagement, **28.2%** on Ad CTR MAPE, **27.9%** on ROAS MAPE, and **97.95%** on webpage likability Pearson $r$. The only exception is memorability, where Henry performs better, though Social Agents still provides clear gains over the baseline. Overall, these results suggest that collective intelligence offers a scalable, model-agnostic path to improving LLM predictions, pointing toward a general paradigm for enhancing both predictive and generative tasks.

To summarize, our paper makes the following three contributions:

**(i) Social Agents Framework.** We introduce *Social Agents*, a multi-agent framework that operationalizes the principle of the Wisdom of Crowds with ensembles of LLM-based human-like persona agents. Each persona agent ensures diversity of opinion, independence, decentralization, and aggregation (Figure 1). We show that Social Agents is *task-agnostic*: evaluated across eleven diverse tasks, it consistently outperforms baselines and trained experts, yielding a 30.5% average improvement on low-level judgment tasks (e.g., webpage likability and ad click-through rate prediction) and a 9.9% average improvement on high-level reasoning tasks such as long-term memorability prediction (see Section 3.2 for results). These gains are robust across decoding temperature settings and aggregation strategies (mean vs. median) (Appendix A.4), indicating that the improvements arise from structured persona diversity in Social Agents rather than stochastic variation in repeated model generations.

**(ii) Model-Agnostic Framework.** Social Agents is model-agnostic, consistently improving performance across both proprietary and open-source LLMs and VLMs of varying scales, rather than relying on gains tied to any specific underlying model architecture. Evaluated across nine different models and 11 diverse tasks, it delivers consistent average improvements of 21.5% over baselines, with clear gains even for smaller parameter-scale models (Appendix A.3.1).

**(iii) Synthetic Data Release.** We release a dataset of persona-conditioned predictions, definitions, and rationales from Social Agents (e.g., ad CTR, webpage likability), capturing how diverse personas and their interactions with digital content are represented across all eleven behavioral tasks.

## 2 SETUP

In this section, we describe how the **Social Agents** framework operates. As illustrated in Fig. 2, a moderator orchestrates the process by first selecting a diverse panel of persona agents from the Agent Factory. Each persona is an LLM agent instantiated with the same backbone model but conditioned on different demographic factors (age, gender, location) and psychographic factors (interests, values, lifestyle). The agents are prompted independently so that there is no interaction or influence among them (see Appendix A.1.1 for the prompting template). When presented with the target query, each agent first generates a rationale from the perspective of its assigned persona and, conditioned on that rationale, then outputs a quantitative prediction. We adopt this rationale-then-score order as an explicit chain-of-thought decision: the rationale grounds the persona's reasoning before the model commits to a numerical value, which we find improves both reproducibility and interpretability of the output. The individual predictions are then aggregated using the wisdom of crowds principle to produce a final ensemble score. For interpretability, the same LLM, configured in a neutral and unconditioned expert mode, synthesizes the set of rationales into a single collective explanation.

### 2.1 MOTIVATION

Social Agents instantiate diverse persona agents, each offering an independent perspective, and aggregate their outputs. Much like the wisdom of crowds, this design captures subjective variability while mitigating bias through ensemble averaging, producing predictions that generalize across domains and approximate population-level responses. Since foundation models are pretrained on diverse human corpora, including social forums like Reddit, they inherently capture a wide range of perspectives (Hu & Collier, 2024; Radford et al., 2019; Gokaslan & Cohen, 2019). By conditioning the same backbone model on different personas, Social Agents samples this distribution of viewpoints rather than relying on a single static prompt, exploiting the model's latent space to simulate psychologically grounded and socially plausible *simulacra* of human responses.

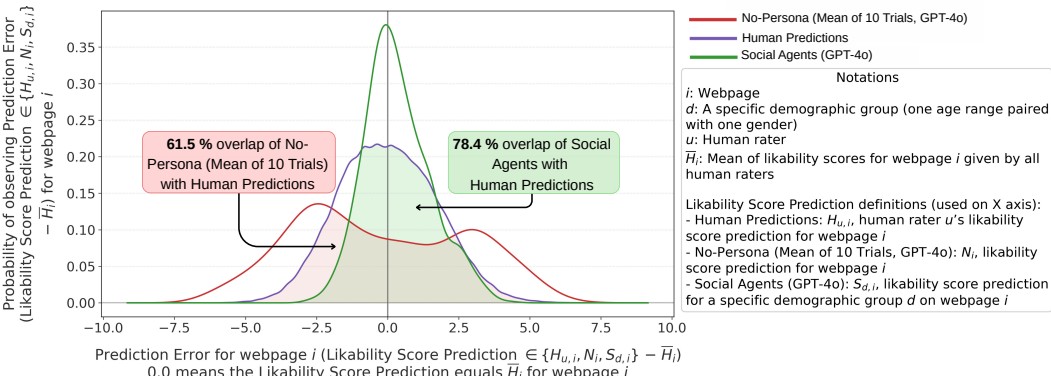

Figure 4: **Comparison of Social Agents (GPT-4o) prediction errors against No-Persona (Mean of 10 Trials, GPT-4o) on the Webpage Likability Prediction task.** Social Agents (GPT-4o), which incorporate individual persona preferences averaged across 10 diverse personas, exhibit greater overlap with the human distribution than the No-Persona (Mean of 10 Trials, GPT-4o) baseline that simply averages 10 calls per webpage. The kernel density estimate (KDE) plot shows the error distributions of Social Agents and the No-Persona (Mean of 10 Trials) baseline relative to the per-webpage human mean prediction $\overline{H_i}$. Shaded areas highlight, for each method, its overlap with the human prediction distribution.

To illustrate, consider predicting the click-through rate (CTR) percentile for the advertisement in Fig. 2. The ad's calm, elegant visuals may resonate with creative professionals but seem less appealing to younger, novelty-seeking users. Social Agents evaluates the ad through multiple personas: for example, a 34-45 year old female marketing graduate with a family predicts 66, a 25-34 year old male marketing graduate 52, a 34-45 year old tech professional 60, an 18-24 year old fashion-conscious female 42, and a 13-17 year old male 66. While these judgments differ, averaging tempers extremes and yields an ensemble score of 54, close to the ground truth of 51. The aggregation is defined as the mean of persona agent predictions, $\hat{S} = \frac{1}{N} \sum_{i=1}^{N} s_i$, where $s_i$ is the score from persona $i$ and $\hat{S}$ is the collective estimate. Unlike classical ensembles that assume independent judgments, persona conditioning induces systematic variation: each agent draws from a distinct yet related distribution, while the shared model backbone introduces correlations across agents. This structure mirrors human crowds, where collective intelligence arises from combining diverse perspectives rather than relying on brute-force repetition of the same model call.

Continuing the same advertisement example, we compare Social Agents to a baseline, *No-Persona (Mean of 10 Trials)*, where the same LLM is queried ten times with the same prompt instead of diverse personas providing different CTR estimates from distinct demographic and psychographic perspectives. This setup mimics a traditional Law of Large Numbers (Bernoulli, 1713) setting: variation comes only from randomness in model sampling, not from systematically different perspectives. Beyond this anecdotal example, we also conduct a systematic experiment comparing Social Agents against the *No-Persona (Mean of 10 Trials)* baseline. As shown in Fig. 4, on the Webpage Likability Prediction task, No-Persona (Mean of 10 Trials) produces a wide distribution of signed errors with only 61.5% KDE overlap with human predictions, whereas Social Agents achieves a tighter error distribution and a higher overlap of 78.4%. The reason is that persona conditioning induces structured variation: different demographic and psychographic profiles yield distinct but complementary distributions, and aggregation cancels bias while preserving diversity. In contrast, repeated calls without personas merely reduce variance and miss the systematic differences in judgment across demographic and psychographic groups, the structured between-group variation, rather than within-group noise, that gives crowd wisdom its predictive power.

Operating under a limited budget, we restrict $N$ to small values, making it appropriate to compare Social Agents with the No-Persona (Mean of 10 Trials) baseline under the same conditions. As shown in the ablation study in Appendix A.3.3, both approaches plateau after about 10-20 calls per datum, indicating diminishing returns beyond this range. Thus, restricting experiments to small $N$ (10 as default in our experiments) offers a fair basis for comparison across tasks.

## 2.2 Tasks & Datasets

To evaluate Social Agents across diverse cognitive domains, we adopt a suite of tasks grounded in Construal Level Theory (CLT) (see Section 1), operationalizing Social Agents through eleven tasks spanning low-, medium-, and high-level construals, capturing how persona agents perceive, engage with, and retain digital content. Full task formulations and task-specific evaluation rules are provided in Appendix A.1.2.

**Low-Construal Tasks.** These tasks involve immediate, surface-level, or perceptual judgments.

**(1) Tweet Engagement Prediction.** Persona agents predict whether a tweet will receive High or Low likes, with outputs aggregated by majority vote (ties re-run). We use the LCBM dataset (Khandelwal et al., 2024), evaluating on 2,339 test tweets. Performance is measured by *accuracy*, the fraction of correct High/Low predictions.

**(2) Ad Click-Through Rate (CTR) Prediction.** Agents estimate the percentile position (0-100) of an ad campaign's CTR relative to other campaigns; CTR captures immediate behavioral outcomes, aligning with low-level construal. We use ad campaigns from Fortune 500 advertisers (3,422 creative industry and 410 real estate campaigns). Evaluation uses three families of metrics: *k-way accuracy*, which discretizes both prediction and ground truth into $k$ equal-width bins (e.g., $k = 3$ or $10$) and counts exact bin matches; *PE@K*, defined as the proportion of predictions whose absolute percentage error lies within $K\%$ of the ground-truth percentile (with $K \in \{10, 20, 30\}$); and *error metrics*, including mean absolute percentage error (MAPE), mean percentage error (MPE) (A.1.2), and root mean squared error (RMSE). Here $k$ (number of bins) and $K$ (percentage tolerance) are independent parameters.

**(3) Webpage Likability Prediction.** Agents rate the likability of a webpage on a 0-10 scale from a single screenshot, capturing immediate aesthetic judgment. We use the Web Aesthetics dataset (Reinecke & Gajos, 2014) of 398 webpage screenshots. Evaluation combines binary *accuracy* (thresholded at 5), error metrics (MAPE, MPE, RMSE), and *Pearson correlation* ($r$) with human ratings.

**(4) Surface Attribute Classification (Topic, Action).** Persona agents classify ads by semantic topic or by identifying explicit actions (e.g., "Buy now"). Both tasks require surface content recognition and observable cues, consistent with low construal. We use the Image and Video Ads dataset (Hussain et al., 2017). Evaluation is based on *accuracy*, computed as exact match with ground-truth labels.

**Medium-Construal Task.** This task involves a future-oriented but still concrete outcome.

**(1) Return on Ad Spend (ROAS) Prediction.** Agents predict the percentile rank of *Return on Ad Spend* (ROAS), defined as the ratio of the revenue a campaign generates to the amount spent running it, where revenue is the dollar value attributed to the campaign and spend is the total media and production cost. ROAS reflects near-term financial consequences, placing it at a medium construal level. We use ad campaigns from Fortune 500 advertisers (3,422 creative industry and 410 real estate campaigns). Evaluation follows the same protocol as CTR: $k$-way accuracy, PE@$K$, and error metrics (MAPE, MPE, RMSE).

**High-Construal Tasks.** These tasks require abstract reasoning, long-term forecasting, or inference of internal states.

**(1) Long-Term Video Memorability Prediction.** Persona agents predict how well video ads will be remembered over time. Because this task involves forecasting retention beyond immediate perception, it reflects high-level construal. We use the LAMBDA dataset (SI et al., 2023), with recall scores rescaled from their original continuous 0-1 range to a 0-10 range for prediction. Evaluation uses *Spearman's rank correlation* ($\rho$), which measures agreement between the rank order of predicted recall scores and human annotations.

**(2) Latent Attribute Classification (Reason, Persuasion, Emotion).** Persona agents identify an ad's underlying justification (Reason), persuasive strategy (Persuasion), or evoked emotion (Emotion); these go beyond surface cues and require deeper semantic reasoning, placing them at high construal. We use the Image and Video Ads dataset (Hussain et al., 2017) for reason and emotion,

and the Video4096 dataset (Bhattacharyya et al., 2023) for persuasion. Evaluation uses *accuracy*, measured as exact match with ground truth.

Although traditionally applied to prediction, we also test whether Social Agents can extend the wisdom of crowds to performative content generation by leveraging diverse social perspectives.

**Tweet Content Generation.** Social Agents generate tweets conditioned on an input advertisement. We use the test set of 2,339 ad-tweet pairs from the LCBM dataset (Khandelwal et al., 2024), which includes 168 million multimodal tweets from 10,135 enterprise accounts (2007–2023) with associated like counts. Each persona agent independently generates a tweet with a supporting rationale. A moderator agent, defined as an expert in tweet generation, synthesizes these into a consolidated tweet that reflects the collective social perspective, along with an aggregated rationale. Evaluation uses *BLEU-N (1-4)* and *ROUGE-1* for lexical overlap with references, and *G-Eval* (Liu et al., 2023), a reference-free evaluator that rates prompt alignment, factual reasonableness, and content suitability.

## 3 RESULTS AND DISCUSSION

### 3.1 EXPERIMENTAL SETUP

To test the model-agnostic nature of Social Agents, we evaluate it on three state-of-the-art LLMs: GPT-4o (Hurst et al., 2024), LLaMA 3.3 70B (Dubey et al., 2024) and Qwen3 32B (Yang et al., 2025). For Webpage Likability Prediction, which requires image understanding, we use their vision-language counterparts: GPT-4o, LLaMA 3.2 90B Vision (Meta AI, 2024) and Qwen2.5 VL 72B (Bai et al., 2025). We also test smaller backbones, including LLaMA 3.1 8B and Qwen3 8B, along with LLaMA 3.2 Vision 11B and Qwen2.5 VL 7B, to evaluate robustness under reduced capacity.

Following the wisdom of crowds principle, our main comparison is between Social Agents and a *No-Persona* variant, where a single LLM expert agent is prompted as a domain specialist. Both are evaluated in a 5-shot setting, except Behavioral Attribute Classification, which uses zero-shot for comparability. For all tasks, we retrieve the top-5 nearest examples (excluding the target item) with OpenAI's `text-embedding-3` model (OpenAI, 2024), using them as few-shot exemplars to ground responses. To ensure a fair comparison, we enforce a strict and uniform 300-token maximum generation budget for both the No-Persona (5-shot) baseline and the Social Agents framework across all tasks and models, so performance gains cannot be attributed to longer outputs but instead to the structured diversity and aggregation mechanism of Social Agents. We further compare Social Agents with task-specific expert models: LCBM (Khandelwal et al., 2024) for Ad CTR, tweet content generation, and engagement prediction; Henry (SI et al., 2023) for memorability; Behavior-LLaVA (Singh et al., 2025) for behavioral attribute classification; and an XGBoost baseline for ROAS and webpage likability prediction. All experiments were run with temperature 0.85 to encourage output diversity.

### 3.2 DISCUSSION

**Social Agents is Model Agnostic.** A key strength of Social Agents is that the benefits of the framework are not tied to a specific model family or scale. Across nine models, both proprietary and open-source, we observe consistent improvements over the No-Persona baseline, averaging 21.5% across all tasks and models (Appendix A.2.1). Larger models such as GPT-4o and Qwen3 32B consistently outperform task-specific experts, with Qwen3 32B achieving the strongest CTR performance and strong ROAS gains, improving PE@20 by 200.0% in the Real Estate domain. Averaged across models, the Real Estate PE@20 improvement is 145.5% (Table 6). In ablation studies, even smaller models such as LLaMA 8B and Qwen 7B, despite lower absolute accuracy, still show clear improvements over their No-Persona counterparts (Appendix A.3.1). This demonstrates that Social Agents provides model-agnostic benefits, with diversity and aggregation yielding consistent gains across scales.

**Performance across prediction and generation tasks.** Social Agents delivers its strongest improvements on low- and medium-level construal tasks, where judgments are immediate and behavior-oriented. On webpage likability prediction, it improves over the No-Persona baseline by 164.2% (Table 5). In Ad CTR prediction, it reduces error by 34.7% (Table 3), while tweet engagement accuracy rises by 21.75% (Table 4). For the medium-level ROAS task, gains are also substantial, with a 27.9% average MAPE reduction and a 75.0% average improvement in PE@20

across both the Creative and Real Estate domains (Table 6). In generative settings, Social Agents improves tweet quality by 21.2% (ROUGE-1), and 7.8% on G-Eval (Table 8). These results suggest that diversity and aggregation provide the greatest benefits for near-term behavioral predictions and creative generation, which, much like human judgments, are easier to assess at lower construal levels (e.g., whether something is likable).

**Performance on high construal level tasks.** On high-level construal tasks requiring abstract reasoning and long-term forecasting, Social Agents produces more moderate but consistent improvements over baselines. In long-term video memorability prediction, it improves over the No-Persona baseline by 13.2% averaged across models (Table 7), with the largest single-model gain of 24.2% coming from GPT-4o. For behavioral attribute classification, Social Agents (GPT-4o) improves over the No-Persona baseline by 11.7% on persuasion and 6.4% on emotion (all labels) (Table 9). Compared with Behavior-LLaVA (zero-shot), gains reach 55.3% on persuasion, 37.6% on action, 34.3% on emotion (all labels), and 25.6% on topic. We note one consistent regression: across all backbone models, Social Agents underperform Behavior-LLaVA (zero-shot) on the clubbed-emotion classification by roughly 22.7%. We attribute this to the coarser binary or few-class structure of clubbed emotion, where Behavior-LLaVA's task-specific finetuning has a larger marginal advantage than persona-driven diversity can recover. This is the only systematic regression we observe across the eleven tasks; on Emotion (All-labels) Social Agents still improves substantially (+6.4% vs. No-Persona, +34.3% vs. Behavior-LLaVA zero-shot).

**Comparison of Social Agents vs trained expert models.** We benchmark Social Agents against task-specific expert models trained on large behavioral datasets. On low- and medium-level tasks, it often surpasses these experts: in Ad CTR prediction it outperforms fine-tuned LCBMs by 34.4% in MAPE reduction (Table 3), while in webpage likability, Social Agents (GPT-4o) exceeds XGBoost by 10.45% on Pearson correlation (Table 5). On ROAS, Social Agents (GPT-4o) exceeds XGBoost on PE@30 by 126.9% in the Creative industry and 18.0% in Real Estate (Table 6). In tweet generation, it also surpasses fine-tuned LCBMs on BLEU-2 and ROUGE-1 (Table 8). Gains are strongest with larger backbones such as GPT-4o, while smaller models provide more modest improvements. High CLT tasks are harder: in long-term memorability, Social Agents improves over No-Persona but does not match Henry, trained on dedicated memorability corpora (Table 7). In behavioral attribute classification, however, it narrows the gap, outperforming Behavior-LLaVA (zero-shot) with gains of up to 55.3% on persuasion, 8.2% on reason, and 34.3% on emotion (all labels) (Table 9). On emotion (clubbed-labels), however, Social Agents underperforms Behavior-LLaVA (zero-shot) by 22.7%(Table 9). These results are achieved with only 5-shot prompting, while expert baselines rely on hundreds of thousands to millions of curated training examples. Overall, Social Agents readily surpasses experts on low- and medium-level tasks, though matching highly specialized models on distant, cognitively demanding ones remains difficult.

**Contribution of Personas in Social Agents.** Figure A.3.3 shows that gains come from persona diversity rather than increasing inference calls. In Ad CTR prediction (Figure 6a), the No-Persona baseline, which averages repeated calls to one prompt, quickly plateaus with higher error. Likewise, the "Wisdom of the Silicon Crowd" Schoenegger et al. (2024), aggregating multiple LLMs without persona conditioning, underperforms compared to a single model with diverse personas. Social Agents achieves lowest MAPE with $N \approx 10 - 20$ personas before diminishing returns. In webpage likability (Figure 6b), the pattern repeats: repeated calls or multi-model ensembles fall short, while just $N = 10$ personas yield the strongest human alignment. This highlights that conditioning on demographic and psychographic profiles provides greater collective intelligence than brute-force scaling.

**Correlation of Social Agents with Human Judgments.** On Webpage Likability Prediction, we measure how well Social Agents' persona-conditioned predictions match the average rating given by human raters of the same demographic group, computed as a Pearson correlation $r$ for each (age, gender) cell over all 398 webpages. Alignment is strong for younger demographics: GPT-4o achieves $r = 0.71$ for 18-24 males and $r = 0.69$ for 18-24 females, and falls steadily with age, dropping to $r = 0.22$-0.25 for the 55+ groups. We posit that LLM pretraining corpora skew toward younger, digitally-native users, so the tastes of older demographics are underrepresented in training and harder to reconstruct via persona conditioning (see Appendix A.3.2 for details). Within the same age bracket, female personas align more closely with human ratings than male personas, and across models GPT-4o leads consistently: weaker alignment with human preferences translates into lower downstream task performance. These results suggest that as LLMs better capture demographic

variation, especially for older cohorts, Social Agents' collective predictions will become even more accurate.

**Effect of temperature across construal levels.** Performance remains stable across decoding temperatures for both high- and low-construal tasks (Tables 17 and 18). On long-term memorability, GPT-4o under Social Agents consistently achieves higher Spearman correlation across temperatures, compared to the No-Persona baseline. Qwen3-32B shows stable performance across temperatures (0.32-0.35), matching or slightly exceeding its No-Persona baseline of 0.33. On CTR prediction, GPT-4o remains around 47.5 percent MAPE across all tested temperatures (47.71, 47.53, 47.60, 47.57), substantially outperforming the 72.45 percent No-Persona baseline. Qwen3-32B shows the same pattern. Across both construal levels, temperature adjustments produce only small numerical variation and do not close the gap between Social Agents and the baseline. The improvements therefore stem from structured persona-level diversity rather than stochastic decoding effects. More details in Section A.8.

## 4 CONCLUSION

We present *Social Agents*, a multi-agent framework that operationalizes the Wisdom of Crowds principle by instantiating a synthetic society of human-like personas whose independent judgments are aggregated into a collective outcome. By using demographic and psychographic attributes to condition each persona, the framework surfaces heterogeneous social perspectives rather than the single uniform response a standalone LLM tends to produce. Across eleven behavioral prediction tasks and nine models, Social Agents consistently improves performance over the No-Persona baseline, averaging 21.5% across all tasks and models. Gains reach 30.5% on low-level judgment tasks and 9.9% on high-level reasoning tasks. Our results position computational crowd simulation as a scalable, interpretable paradigm for behavioral prediction and content generation, opening a path to using LLMs as proxy populations for studying how diverse human groups respond to digital content.

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

## A  APPENDIX

### A.1  EXPERIMENTAL DETAILS

#### A.1.1  PROMPTING PERSONAS IN SOCIAL AGENTS

We use structured prompts that elicit nuanced, persona-driven judgments. Each prompt grounds the persona agent in a specific demographic-psychographic profile, provides context, and defines the evaluation task. We can define the complete prompt $P_i$ for an agent $i$ as a structured concatenation of four key components:

$$P_i = \text{System}(\mathcal{S}) \oplus \text{Persona}(\mathcal{D}_i) \oplus \text{Task}(\mathcal{C}, \mathcal{T}) \oplus \text{Format}(\mathcal{F})$$

where $\mathcal{S}$ is a system-level instruction, $\mathcal{D}_i$ is the unique demographic profile for agent $i$, $\mathcal{C}$ is the task context including few-shot examples, $\mathcal{T}$ is the specific task goal, $\mathcal{F}$ is the required output format, and $\oplus$ denotes string concatenation.

The general structure of our prompts is encapsulated in the template shown below. We can use this structure to adapt Social Agents to the rest of the eleven behavioral tasks by simply modifying the task-specific components while keeping the core persona agent descriptions consistent.

> **Prompt Structure**
>
> **Persona Description** ($\mathcal{D}_i$): You are a `[Demographic Attributes]`. `[A detailed, narrative description of the persona's values, media consumption habits, cultural touchstones, and what typically influences their judgments and preferences.]`
>
> **Task Context & Goal** ($\mathcal{C}, \mathcal{T}$): You are given 5 examples `[stimuli]` and their corresponding `[metric]` scores (on a 0-100 or 0-10 scale). You are now shown a new `[stimulus]`. Your task is to judge how `[target attribute]` this `[stimulus]` is likely to be for `[your demographic group]`. `[Further clarification on what cognitive or emotional factors to consider for the judgment.]`
>
> **Output Format** ($\mathcal{F}$): Return: Reason: `[Provide a concise explanation for your judgment, rooted in your persona's perspective.]` Answer: `[A numerical score in the specified range, e.g., 0-100 or 0-10, based on the task.]` ← You must include this score.

For our experiments, we instantiate a panel of 10 distinct personas based on a cross-section of age and gender. These personas, summarized in Table 2, are used across all evaluation tasks. The only exception is for the Long-Term Video Memorability Prediction task. To align with the demographics of the original dataset's human annotators, we use a more focused panel consisting of only the first four personas: 18-24 female, 18-24 male, 25-34 female and 25-34 male.

#### A.1.2  DETAILED TASKS AND METRIC FORMULATION

We provide the mathematical formulations for the evaluation metrics used in our study. Throughout, an item refers to an ad, webpage, or video depending on the task. Predictions are produced by the system under evaluation (e.g., our Social Agents, or the No-Persona baseline). For Social Agents, persona outputs are aggregated by majority vote for categorical predictions in the Tweet Engagement Prediction and Behavioral Attribute Classification tasks, while for rest of the tasks, the predictions

| Task Name | Construal Level | Construal Level Justification | Metrics Used |
|---|---|---|---|
| Tweet Engagement Prediction | Low | Judging likes from surface tweet cues; fast, low-abstraction decision. | Accuracy |
| Ad Click-Through Rate (CTR) Prediction | Low | Predicts immediate clicks based on perceptual and short-horizon content cues. | $k$-way Accuracy, PE@$K$, MAPE, MPE, RMSE |
| Webpage Likability Prediction | Low | Judgments rely on immediate perceptual features (e.g., color, layout, and visual hierarchy); requires rapid, surface-level evaluation. | Pearson Correlation, Accuracy, MAPE, MPE, RMSE |
| Behavioral Attribute Classification - Surface (Topic) | Low | Topic can be identified directly from explicit video ad text or visuals, without abstraction or long-term reasoning. | Accuracy |
| Behavioral Attribute Classification - Surface (Action) | Low | Detects explicit calls-to-action (e.g., "Buy now") that are visible on the surface of the video ad; minimal inference required. | Accuracy |
| Return on Ad Spend (ROAS) Prediction | Medium | Involves forecasting near-term financial outcomes; requires reasoning about consumer response and cost effectiveness beyond immediate perceptual cues. | $k$-way Accuracy, PE@$K$, MAPE, MPE, RMSE |
| Long-Term Video Memorability Prediction | High | Requires predicting whether content will be recalled weeks later; demands abstract reasoning about memory processes, semantic associations, and persuasive impact. | Spearman Correlation |
| Behavioral Attribute Classification - Latent (Reason, Persuasion, Emotion) | High | Goes beyond surface features to infer latent intent, persuasive strategies, and emotional resonance, which require deeper abstraction. | Accuracy |
| Tweet Content Generation | — | — | BLEU-$N$ (1–4), ROUGE-1, G-Eval |

Table 1: Tasks, their construal levels, justifications, and evaluation metrics. Low-level tasks focus on immediate, perceptual judgments; medium-level tasks address near-term outcomes; and high-level tasks involve abstract reasoning or long-horizon forecasting. The *tweet content generation* task is not associated with construal levels, as it involves producing performative content or text rather than making predictive judgments.

are averaged using an aggregator before evaluation. The full mapping of tasks to their construal levels with justification and evaluation metrics is summarized in Table 1.

**Accuracy:** Accuracy measures the fraction of items (ads, webpages, or videos) for which the system's final predicted label matches the human-provided label. It is defined as:

$$\text{Accuracy} = \frac{1}{n} \sum_{i=1}^{n} \mathbf{1}[y_i = \hat{y}_i],$$

where $y_i$ is the human label (e.g., High or Low engagement for a tweet, High or Low likability score for a webpage, or a behavioral attribute for a video) and $\hat{y}_i$ is the system's prediction for item $i$.

For Webpage Likability Prediction, the averaged predicted likability score (0-10) is converted into a binary label using a threshold of 5 (High if score $> 5$, Low if $\leq 5$). This predicted category is then compared against the human-rated category. For Tweet Engagement Prediction, each persona agent outputs a binary label (High or Low engagement), and the final system prediction is obtained by majority voting across personas. In case of ties in Social Agents, the framework is re-run for that tweet to resolve the decision. For Behavioral Attribute Classification on videos, the prediction is correct only if the chosen attribute label (one of Topic, Action, Reason, Persuasion, or Emotion) exactly matches one of the five human annotated label.

For discretized continuous prediction tasks on ads, such as CTR or ROAS percentile estimation, we also report $k$-way accuracy. In this setting, both the ground-truth percentile and the predicted percentile are discretized into $k$ equal-width bins (e.g., $k = 3$ or $k = 10$). A prediction is counted correct if the predicted bin matches the actual bin:

$$\text{Accuracy}_{k\text{-way}} = \frac{1}{n} \sum_{i=1}^{n} \mathbf{1}\big[\text{bin}(A_i) = \text{bin}(P_i)\big],$$

where $A_i$ is the actual percentile and $P_i$ is the predicted percentile for item $i$.

**BLEU and ROUGE (Papineni et al., 2002; Lin, 2004)** For the tweet content generation task, BLEU-$N$ measures $n$-gram overlap precision between a generated tweet and a human reference tweet, combined with a brevity penalty to avoid favoring short outputs:

$$\text{BLEU} = \text{BP} \cdot \exp\Big( \sum_{n=1}^{N} w_n \log p_n \Big), \qquad \text{BP} = \begin{cases} 1 & c > r \\ \exp(1 - r/c) & c \leq r \end{cases}$$

where $p_n$ is the modified precision for $n$-grams and $c, r$ are the predicted and reference lengths. We report BLEU-1 through BLEU-4. ROUGE-1 reports unigram overlap between generated and reference tweets, with precision, recall, and F1 score. Standard normalization (lowercasing, punctuation stripping, tokenization) is applied.

**Mean Absolute Percentage Error (MAPE) and Mean Percentage Error (MPE):** These metrics are used for numerical prediction tasks such as Ad Click-Through Rate (CTR) percentile, Return on Ad Spend (ROAS) percentile, and Webpage Likability Prediction. They capture the average proportional error magnitude (MAPE) and the average signed bias (MPE):

$$\text{MAPE} = \frac{1}{n} \sum_{i=1}^{n} \left| \frac{A_i - P_i}{A_i} \right| \times 100\%, \quad \text{MPE} = \frac{1}{n} \sum_{i=1}^{n} \left( \frac{A_i - P_i}{A_i} \right) \times 100\%.$$

Here, $A_i$ is the human-provided ground-truth value for the $i^{\text{th}}$ item (e.g., actual CTR percentile, or actual ROAS percentile), and $P_i$ is the system's predicted value for the same item. MAPE reports the average absolute percentage deviation, while MPE indicates whether the predictions systematically overestimate (positive) or underestimate (negative) relative to human ground truth.

**Root Mean Squared Error (RMSE):** Used in the webpage likability prediction task, RMSE summarizes the magnitude of prediction errors by penalizing larger deviations more strongly:

$$\text{RMSE} = \sqrt{\frac{1}{n} \sum_{i=1}^{n} (P_i - A_i)^2}.$$

Here, $n$ is the total number of items, $A_i$ is the human-provided ground-truth value for the $i^{\text{th}}$ item, and $P_i$ is the system's predicted value for that item.

**Pearson Correlation Coefficient ($r$):** For webpage likability prediction task, Pearson's $r$ measures the strength of linear association between system predictions and human provided ground truth across webpages:

$$r = \frac{\sum_{i=1}^{n} (A_i - \bar{A})(P_i - \bar{P})}{\sqrt{\sum_{i=1}^{n} (A_i - \bar{A})^2} \sqrt{\sum_{i=1}^{n} (P_i - \bar{P})^2}}, \quad \bar{A} = \frac{1}{n} \sum_{i=1}^{n} A_i, \quad \bar{P} = \frac{1}{n} \sum_{i=1}^{n} P_i.$$

Here, $A_i$ is the human-provided ground-truth value for the $i^{\text{th}}$ item, and $P_i$ is the corresponding system-predicted value for that same item. $\bar{A}$ is the mean of all human-provided ground-truth values across the webpages, and $\bar{P}$ is the mean of all system-predicted values across the webpages.

**Spearman Rank Correlation ($\rho$):** For long-term video memorability prediction, Spearman's $\rho$ measures whether the ranking of system-predicted recall scores aligns with the ranking of human-measured recall scores for the videos:

$$\rho = 1 - \frac{6 \sum_{i=1}^{n} d_i^2}{n(n^2 - 1)}, \quad d_i = \text{rank}(A_i) - \text{rank}(P_i).$$

Here, $A_i$ is the human recall score, $P_i$ is the predicted recall score by the system, and $d_i$ is the difference between the ranks of these two scores for the $i^{\text{th}}$ video.

**Prediction Error at $K$ (PE@$K$):** For ad CTR and ROAS percentile prediction, PE@$K$ measures the percentage of ads whose absolute percentage error falls within a tolerance of $K\%$ of the ground-truth percentile:

$$\text{PE@}K = \frac{1}{n} \sum_{i=1}^{n} \mathbf{1} \left[ \left| \frac{P_i - A_i}{A_i} \right| \times 100 \leq K \right],$$

with $K \in \{10, 20, 30\}$. Here, $A_i$ is the human-provided ground-truth percentile for the $i^{\text{th}}$ ad, $P_i$ is the predicted percentile for that ad, and the indicator function counts an ad as correct if its prediction error is within the specified tolerance $K$.

**G-Eval (Liu et al., 2023):** For the tweet content generation task, G-Eval uses a large language model (GPT-4o) as an evaluator. It takes three inputs: the original ad prompt, the system-generated tweet, and one human reference tweet. Guided by a rubric, the evaluator assigns scores along three dimensions: **(i) Prompt Alignment:** whether the generated tweet effectively addresses the intent or theme of the target context (i.e., the ad's intent, theme, or message), even if phrased differently from the human reference tweet; **(ii) Factual Reasonableness:** whether the generated tweet is factually accurate or at least plausible given the target context (i.e., the ad's intent, theme, or message),

| Demographic Profile | Key Psychographic Traits |
|---|---|
| 18-24 female | Values bold aesthetics, authenticity, and viral social media trends. |
| 18-24 male | Attuned to humor, shock value, memes, and gaming culture. |
| 25-34 female | Values aesthetics, emotional resonance, and lifestyle relevance. |
| 25-34 male | Values clarity, style, and visuals related to aspiration and technology. |
| 35-44 female | Appreciates authenticity, emotional intelligence, and meaningful content. |
| 35-44 male | Responds to visuals that are smart, grounded, and purpose-driven. |
| 45-54 female | Drawn to emotional clarity, purpose, and community-oriented themes. |
| 45-54 male | Appreciates thoughtful, honest visuals grounded in wisdom and reality. |
| 55+ female | Resonates with warmth, purpose, legacy, and deep emotional connection. |
| 55+ male | Values sincerity, legacy, and visuals with clear, meaningful value. |

Table 2: **Demographic Profiles and Psychographic Traits of the 10 Social Agent Personas.** Each persona combines a demographic group (age group paired with a gender) with its corresponding key psychographic traits. The table highlights the types of aesthetics, values, and content themes that each group is most likely to prefer when evaluating or responding to an ad, webpage, or video.

avoiding clear inaccuracies; and **(iii) Content Suitability:** whether the generated tweet is coherent, well-structured, clear, and appropriate for the Twitter platform in terms of tone and brevity.

Each dimension is scored separately, and the final G-Eval score is the average of the three sub-scores:

$$\text{G-Eval Score} = \tfrac{1}{3}\left(\text{Score}_{\text{prompt\_alignment}} + \text{Score}_{\text{factual\_reasonableness}} + \text{Score}_{\text{content\_suitability}}\right).$$

This setup allows the evaluation to capture not only word overlap but also the overall quality and contextual appropriateness of the generated tweets.

## A.2 EXPERIMENTAL RESULTS

### A.2.1 PERFORMANCE OF SOCIAL AGENTS

Tables 3, 4, 6, 5, 7, 8 and 9 summarize results across all eleven behavioral prediction tasks. GPT-4o, as the strongest model, consistently outperforms both baselines and trained experts, whereas all models show consistent gains. By contrast, LLaMA, with weaker alignment to human preferences, struggles to match the performance of specialized expert models.

| Model | Accuracy (%) ↑ |
|---|---|
| GPT-3.5 (OpenAI, 2023) | 61.52 |
| LCBM (Finetuned on Twitter Data) (Khandelwal et al., 2024) | 85.99 |
| LCBM (Finetuned on Twitter + Youtube Data) (Khandelwal et al., 2024) | 84.53 |
| No-Persona (GPT-4o) | 70.27 |
| Social Agents (GPT-4o) | 86.90 |
| No-Persona (Qwen3 32B) | 54.20 |
| Social Agents (Qwen3 32B) | 63.85 |
| No-Persona (LLaMA 3.3 70B) | 51.50 |
| Social Agents (LLaMA 3.3 70B) | 63.75 |
| **Improvement of Social Agents over No-Persona** | 21.75% |

Table 4: **Tweet Engagement Prediction.** Performance evaluated using Accuracy (%, higher is better). Social Agents outperform the No-Persona baseline across all models, with individual relative improvements of 23.68% (GPT-4o), 17.78% (Qwen3 32B), and 23.79% (LLaMA 3.3 70B), yielding a mean relative improvement of 21.75% across the three backbone models. Our approach achieves performance closely comparable to that of fine-tuned LCBMs, with Social Agents (GPT-4o) surpassing the average LCBM performance by 1.92% (averaged across both variants). Positive gains are shown in green. Best models are denoted in green, and runner-ups in blue.

| Model & Dataset | MAPE ↓ | 3-way accuracy (%) ↑ | 10-way accuracy (%) ↑ |
|---|---|---|---|
| **Creative & Real Estate Industries** | | | |
| Random Baseline | 111.03 | 33.33 | 10.00 |
| **Creative Industry** | | | |
| XGBoost (Chen & Guestrin, 2016) | 55.40 | 48.82 | 22.62 |
| LCBM (Zero-shot) (Khandelwal et al., 2024) | 80.21 | 71.75 | 43.90 |
| LCBM (Finetuned) (Khandelwal et al., 2024) | 75.67 | 70.68 | 42.77 |
| No-Persona (LLaMA 3.3 70B) | 85.80 | 53.97 | 29.96 |
| Social Agents (LLaMA 3.3 70B) | 62.44 | 69.00 | 25.20 |
| No-Persona (Qwen3 32B) | 59.13 | 70.40 | 39.00 |
| Social Agents (Qwen3 32B) | 43.31 | 75.70 | 29.60 |
| No-Persona (GPT-4o) | 72.45 | 40.04 | 18.91 |
| Social Agents (GPT-4o) | 47.60 | 77.19 | 45.79 |
| **Improvement of Social Agents over No-Persona** | 29.43% | 42.72% | 34.05% |
| **Improvement of Social Agents over LCBM** | 34.36% | 3.09% | -23.62% |
| **Real Estate Industry** | | | |
| XGBoost (Chen & Guestrin, 2016) | 48.90 | 60.92 | 24.27 |
| No-Persona (Qwen3 32B) | 63.36 | 60.00 | 26.10 |
| Social Agents (Qwen3 32B) | 50.79 | 65.90 | 23.40 |
| No-Persona (LLaMA 3.3 70B) | 75.14 | 63.40 | 27.30 |
| Social Agents (LLaMA 3.3 70B) | 55.59 | 55.60 | 19.30 |
| No-Persona (GPT-4o) | 70.27 | 61.50 | 23.20 |
| Social Agents (GPT-4o) | 45.62 | 69.40 | 29.60 |
| **Improvement of Social Agents over No-Persona** | 26.98% | 3.46% | -4.02% |

Table 3: **Ad Click-Through Rate (CTR) Prediction.** Results on datasets we constructed from Creative and Real Estate industries listed in the Forbes Fortune 500, evaluated using **MAPE** (lower is better) and **3-way and 10-way accuracy** (higher is better). Social Agents reduce prediction error compared to No-Persona baselines, with average MAPE reductions of 26.6% (LLaMA 3.3 70B), 23.3% (Qwen3 32B), and 34.7% (GPT-4o) across both industries. Compared to LCBM, Social Agents achieve a further 34.36% lower MAPE in the creative industry. We report the average improvements across all metrics: Social Agents compared to No-Persona (averaged over all models) and Social Agents compared to LCBM (averaged over zero-shot and fine-tuned models). Positive gains are shown in green. Best models are highlighted in green, and runner-ups in blue. We compare with a moving average baseline, which serves as our random baseline.

| Model | Pearson Correlation (r) ↑ | MPE (%) ↓ | RMSE ↓ | Accuracy (%) ↑ |
|---|---|---|---|---|
| XGBoost | 0.67 | 19.70 | 0.72 | 77.55 |
| No-Persona (LLaMA 3.2 90B Vision) | 0.35 | 36.55 | 1.98 | 57.73 |
| Social Agents (LLaMA 3.2 90B Vision) | 0.61 | 25.94 | 1.31 | 56.70 |
| No-Persona (Qwen2.5 VL 72B) | 0.38 | 21.25 | 1.15 | 70.10 |
| Social Agents (Qwen2.5 VL 72B) | 0.59 | 19.69 | 1.03 | 73.20 |
| No-Persona (GPT-4o) | 0.28 | 59.31 | 2.86 | 61.22 |
| Social Agents (GPT-4o) | 0.74 | 14.95 | 0.84 | 80.61 |
| **Improvement of Social Agents over No-Persona** | 97.95% | 37.05% | 38.30% | 11.44% |

Table 5: **Webpage Likability Prediction.** Performance evaluated using Pearson correlation (r, higher is better), Mean Percentage Error (MPE, lower is better), Root Mean Squared Error (RMSE, lower is better), and Accuracy (%, higher is better). Social Agents outperform the No-Persona baseline across all models in Pearson Correlation, with individual improvements of 74.29% (LLaMA 3.2 90B Vision), 55.26% (Qwen2.5 VL 72B), and 164.2% (GPT-4o). Overall improvement percentages represent the relative improvement of Social Agents over No-Persona (averaged across all models) for each metric. Positive gains are shown in green. Best models are denoted in green, and runner-ups in blue.

| Model & Dataset | MAPE ↓ | PE@10 ↑ | PE@20 ↑ | PE@30 ↑ |
|---|---|---|---|---|
| **Creative & Real Estate Industries** | | | | |
| Human Baseline | 104.70 | 0.071 | 0.176 | 0.307 |
| **Creative Industry** | | | | |
| XGBoost (Chen & Guestrin, 2016) | 52.12 | 0.08 | 0.17 | 0.26 |
| No-Persona (LLaMA 3.3 70B) | 81.15 | 0.20 | 0.31 | 0.42 |
| Social Agents (LLaMA 3.3 70B) | 74.57 | 0.12 | 0.24 | 0.34 |
| No-Persona (Qwen3 32B) | 52.90 | 0.18 | 0.36 | 0.47 |
| Social Agents (Qwen3 32B) | 43.57 | 0.16 | 0.32 | 0.45 |
| No-Persona (GPT-4o) | 75.70 | 0.16 | 0.32 | 0.42 |
| Social Agents (GPT-4o) | 58.76 | 0.27 | 0.47 | 0.59 |
| **Improvement of Social Agents over No-Persona** | 16.04% | 5.88% | 4.39% | 5.1% |
| **Real Estate Industry** | | | | |
| XGBoost (Chen & Guestrin, 2016) | 48.63 | 0.13 | 0.26 | 0.39 |
| No-Persona (LLaMA 3.3 70B) | 134.13 | 0.03 | 0.09 | 0.12 |
| Social Agents (LLaMA 3.3 70B) | 125.19 | 0.06 | 0.11 | 0.17 |
| No-Persona (Qwen3 32B) | 105.91 | 0.02 | 0.03 | 0.04 |
| Social Agents (Qwen3 32B) | 80.21 | 0.04 | 0.09 | 0.15 |
| No-Persona (GPT-4o) | 199.93 | 0.06 | 0.10 | 0.14 |
| Social Agents (GPT-4o) | 50.23 | 0.15 | 0.32 | 0.46 |
| **Improvement of Social Agents over No-Persona** | 39.8% | 136.4% | 145.5% | 128.6% |

Table 6: **Return on Ad Spend (ROAS) Prediction.** Results on datasets we constructed from Creative and Real Estate industries listed in the Forbes Fortune 500, evaluated using **MAPE** (lower is better) and **PE@K** (higher is better). Social Agents reduce prediction error compared to No-Persona baselines in most settings, with average MAPE reductions of 7.4% (LLaMA 3.3 70B), 20.95% (Qwen3 32B), and 48.65% (GPT-4o) across both industries. We report the average improvements across all metrics: Social Agents compared to No-Persona (averaged over all models). Positive gains are shown in green. Best models are denoted in green, and runner-ups in blue.

| Model | Spearman Correlation ($\rho$) ↑ |
|---|---|
| Human Consistency | 0.55 |
| 10-shot GPT-3.5 (OpenAI, 2023) | 0.06 |
| Regression using ViT feats (ViTMem) (Hagen & Espeseth, 2023) | 0.08 |
| Henry trained on individual datasets (SI et al., 2023) | 0.55 |
| Henry trained on all (combined) datasets (SI et al., 2023) | 0.52 |
| No-Persona (LLaMA 3.3 70B) | 0.26 |
| Social Agents (LLaMA 3.3 70B) | 0.30 |
| No-Persona (Qwen3 32B) | 0.33 |
| Social Agents (Qwen3 32B) | 0.33 |
| No-Persona (GPT-4o) | 0.33 |
| Social Agents (GPT-4o) | 0.41 |
| **Improvement of Social Agents over No-Persona** | 13.2% |

Table 7: **Long-Term Video Memorability (LAMBDA) Prediction.** Performance evaluated using Spearman correlation ($\rho$, higher is better). Among non-trained methods, Social Agents (GPT-4o) shows the best performance, highlighted in yellow. Social Agents outperform No-Persona baselines with individual improvements of 15.38% (LLaMA 3.3 70B), 0.00% (Qwen3 32B), and 24.24% (GPT-4o). We report the average improvements: Social Agents compared to No-Persona (averaged over all models). Positive gains are shown in green and no improvement in gray. Best models are denoted in green, and runner-ups in blue.

| Model | BLEU-1 ↑ | BLEU-2 ↑ | BLEU-3 ↑ | BLEU-4 ↑ | ROUGE-1 ↑ | G-Eval ↑ |
|---|---|---|---|---|---|---|
| GPT-3.5 (ICL) (OpenAI, 2023) | 55.82 | 43.74 | 32.76 | 24.90 | 15.26 | - |
| LCBM (Finetuned on Twitter Data) (Khandelwal et al., 2024) | 66.15 | 50.50 | 37.71 | 25.95 | 15.91 | - |
| LCBM (Finetuned on Twitter + YouTube Data) (Khandelwal et al., 2024) | 67.26 | 51.32 | 38.35 | 29.59 | 16.14 | - |
| No-Persona (GPT-4o) | 62.41 | 47.33 | 32.90 | 23.77 | 14.57 | 0.57 |
| Social Agents (GPT-4o) | 70.31 | 54.54 | 40.38 | 31.71 | 14.89 | 0.61 |
| No-Persona (Qwen3 32B) | 64.98 | 48.14 | 34.45 | 25.60 | 9.39 | 0.53 |
| Social Agents (Qwen3 32B) | 66.43 | 49.73 | 36.10 | 27.12 | 14.01 | 0.57 |
| No-Persona (LLaMA 3.3 70B) | 49.58 | 36.16 | 25.48 | 18.88 | 6.98 | 0.46 |
| Social Agents (LLaMA 3.3 70B) | 55.87 | 40.70 | 28.64 | 21.01 | 7.83 | 0.50 |
| **Improvement of Social Agents over No-Persona** | 9.19% | 10.36% | 13.31% | 16.87% | 21.19% | 7.76% |

Table 8: **Tweet Content Generation.** Results across models measured via BLEU (1-4), ROUGE-1, and G-Eval metrics (higher is better). Social Agents outperform No-Persona baselines across all models, with individual improvements of 33.40% (GPT-4o), 5.94% (Qwen3 32B), and 11.28% (LLaMA 3.3 70B) for BLEU-4; 2.20% (GPT-4o), 49.20% (Qwen3 32B), and 12.18% (LLaMA 3.3 70B) for ROUGE-1. Compared to the averaged LCBM baselines across both fine-tuned variants, Social Agents (GPT-4o) achieves a 7.13% improvement on BLEU-2 and a 14.19% improvement on BLEU-4, while LCBM remains stronger on BLEU-1 and ROUGE-1. Overall improvement percentages represent the relative improvement of Social Agents over No-Persona (averaged across all models) for each metric. Best performing models are highlighted in green , and runner-ups in blue .

| Model | Topic | Emotion | | Persuasion | Action | Reason |
|---|---|---|---|---|---|---|
| | | **All labels** | **Clubbed** | | | |
| Random | 2.63 | 3.37 | 14.3 | 8.37 | 3.34 | 3.34 |
| Hussain et al. (Finetuned) (Hussain et al., 2017) | 35.1 | 32.8 | - | - | - | 48.45 |
| Intern-Video (Finetuned) (Wang et al., 2022) | 57.47 | 36.08 | 86.59 | 5.47 | 6.8 | 7.1 |
| VideoChat (Zero-shot) (Li et al., 2023) | 9.07 | 3.09 | 5.1 | 10.28 | - | - |
| Video4096 (GPT-3.5 Generated Story + GPT-3.5 Classifier) (Bhattacharyya et al., 2023) | 51.6 | 11.68 | 79.69 | 35.02 | 66.27 | 59.59 |
| Video4096 (GPT-3.5 Generated Story + Flan-t5-xxl Classifier) (Bhattacharyya et al., 2023) | 60.5 | 10.8 | 79.10 | 33.41 | 79.22 | 81.72 |
| Video4096 (Vicuna Generated Story + Flan-t5-xxl Classifier) (Bhattacharyya et al., 2023) | 57.38 | 9.8 | 76.60 | 30.11 | 77.38 | 80.66 |
| Video4096 (Vicuna Generated Story + Roberta Classifier, Finetuned) (Bhattacharyya et al., 2023) | 71.3 | 33.02 | 84.20 | 64.67 | 42.96 | 39.09 |
| Behavior-LLaVA (Zero-shot, w/ video + GPT-3.5 story) (Singh et al., 2025) | 60.09 | 12.84 | 79.94 | 36.12 | 67.10 | 79.18 |
| Behavior-LLaVA (Finetuned, w/ video + GPT-3.5 story) (Singh et al., 2025) | 71.2 | 39.55 | 86.17 | 65.03 | 80.44 | 81.67 |
| No-Persona (GPT-4o + GPT-4o Story) | 70.76 | 16.21 | 61.53 | 50.20 | 91.20 | 80.75 |
| Social Agents (GPT-4o + GPT-4o Story) | 75.47 | 17.24 | 61.81 | 56.09 | 92.30 | 85.71 |
| **Improvement of Social Agents over No-Persona** | 6.7% | 6.4% | 0.5% | 11.7% | 1.2% | 6.1% |
| **Improvement of Social Agents over Behavior-LLaVA (Zero-shot)** | 25.6% | 34.3% | -22.7% | 55.3% | 37.6% | 8.2% |
| No-Persona (Qwen3 32B + GPT-4o Story) | 67.85 | 16.01 | 63.01 | 41.72 | 88.90 | 80.75 |
| Social Agents (Qwen3 32B + GPT-4o Story) | 69.45 | 15.96 | 61.71 | 52.89 | 90.30 | 83.95 |
| **Improvement of Social Agents over No-Persona** | 2.4% | -0.3% | -2.0% | 26.8% | 1.6% | 4.0% |
| **Improvement of Social Agents over Behavior-LLaVA (Zero-shot)** | 15.6% | 24.3% | -22.8% | 46.4% | 34.6% | 6.0% |
| No-Persona (LLaMA 3.3 70B + GPT-4o Story) | 70.11 | 18.11 | 61.67 | 46.31 | 88.70 | 81.80 |
| Social Agents (LLaMA 3.3 70B + GPT-4o Story) | 70.86 | 17.21 | 62.03 | 53.49 | 88.80 | 88.40 |
| **Improvement of Social Agents over No-Persona** | 1.1% | -5.0% | 0.6% | 15.5% | 0.1% | 8.1% |
| **Improvement of Social Agents over Behavior-LLaVA (Zero-shot)** | 17.9% | 34.0% | -22.40% | 48.1% | 32.3% | 11.6% |

Table 9: **Behavioral Attribute Classification.** Performance evaluated using Accuracy (%, higher is better) across five dimensions: Topic, Emotion (All-labels and Clubbed), Persuasion, Action, and Reason. Social Agents improve over No-Persona baselines by up to 26.8% (Action, Qwen3-32B), and outperform Behavior-LLaVA (Zero-shot) by up to 55.3% on Persuasion. We observe one systematic regression: across all backbones, Social Agents underperform Behavior-LLaVA (Zero-shot) on Emotion (Clubbed) by roughly 22.7% (see Section 3.2). Positive gains are highlighted in green, declines in red. Best models are marked in green , runner-ups in blue .

## A.3 Ablation Results

### A.3.1 Evaluation of Social Agents on Lower Parameter Models

To assess whether compact, lower-parameter models can still capture the human preference signals required for Social Agents to function effectively, we evaluate such models on the Webpage Likability Prediction and Long-Term Video Memorability Prediction tasks (Table 10, 11).

| Model and Method | Pearson Correlation ↑ | MPE (%) ↓ | RMSE ↓ | Accuracy (%) ↑ |
|---|---|---|---|---|
| **LLaMA 3.2 11B Vision** | | | | |
| No-Persona | 0.02 | 49.21 | 2.50 | 60.22 |
| No-Persona (Mean of 10 Trials) | 0.37 | 41.44 | 1.81 | 40.21 |
| Social Agents | 0.38 | 27.47 | 1.21 | 60.82 |
| **Qwen2.5 VL 7B** | | | | |
| No-Persona | 0.54 | 47.84 | 2.12 | 41.24 |
| No-Persona (Mean of 10 Trials) | 0.54 | 26.22 | 1.33 | 68.42 |
| Social Agents | 0.58 | 27.54 | 1.34 | 63.27 |

Table 10: **Performance comparison on the Webpage Likability Prediction task using lower-parameter models.** Metrics reported include Pearson Correlation (↑), Mean Percentage Error (MPE) (↓), Root Mean Squared Error (RMSE) (↓), and Accuracy (↑). Qwen2.5 VL 7B with Social Agents outperforms all other configurations across all metrics, achieving the highest correlation and accuracy, and the lowest MPE and RMSE. For LLaMA 3.2 11B Vision, we find that the No-Persona (Mean of 10 Trials) baseline performs better than Social Agents, highlighting the limitation of these smaller models in understanding personas. Best-performing models are highlighted in green, and runner-ups in blue.

| Model and Method | Spearman Correlation ↑ |
|---|---|
| **LLaMA 3.1 8B** | |
| No-Persona | 0.08 |
| No-Persona (Mean of 10 Trials) | 0.12 |
| Social Agents | 0.12 |
| **Qwen3 8B** | |
| No-Persona | 0.21 |
| No-Persona (Mean of 10 Trials) | 0.19 |
| Social Agents | 0.26 |

Table 11: **Performance comparison on the Long-Term Video Memorability task using lower-parameter models.** Metrics reported include Spearman correlation (↑). Qwen3 8B with Social Agents achieves the highest rank-order correlation with human judgments. For LLaMA 3.1 8B, we observe that the No-Persona (Mean of 10 Trials) configuration performs comparably to the Social Agents variant, highlighting the model's limited ability to interpret personas. Best-performing strategies are highlighted in green, and runner-ups in blue.

In the Webpage Likability Prediction task, Qwen2.5 VL 7B with Social Agents achieves a 7% relative gain in correlation, a 42% reduction in mean percentage error (MPE), and a 53% uplift in accuracy. LLaMA 3.2 11B Vision shows the largest error reduction, with a 44% drop in MPE and a 52% reduction in RMSE when using Social Agents compared to the No-Persona configuration. However, its best absolute performance in correlation and accuracy is obtained by the No-Persona (Mean of 10 Trials) baseline rather than Social Agents, suggesting that some smaller models benefit primarily in error reduction while missing finer gains in discrimination and alignment with human judgments.

In the Long-Term Video Memorability Prediction task, Qwen3 8B with Social Agents improves performance by 24% over the No-Persona baseline, while LLaMA 3.1 8B yields similar moderate gains across both No-Persona (Mean of 10 Trials) and Social Agents configurations.

Overall, Social Agents generally outperform the No-Persona baseline, though the magnitude of improvement varies. Gains tend to be smaller or inconsistent in lower-capacity models, reflecting limits in model expressivity and reduced exposure to diverse human data. Even so, Social Agents help narrow the performance gap to larger models by reducing prediction errors.

### A.3.2 Alignment of Social Agents' Predictions with Human Judgments

To measure how well simulated persona agents in the Social Agents framework reproduce real human preferences, we use the Webpage Likability Prediction task (Figure 5). The dataset contains 398 webpages, each rated on a 10-point Likert scale by nearly 40,000 participants from 179 countries, providing diverse demographic coverage. For evaluation, we compute the Pearson correlation ($r$) between the likability scores predicted by Social Agents persona demographic groups and the

average human ratings from the corresponding demographic groups for each webpage. Results are reported across three different model backbones.

Figure 5: **Correlation between Social Agents' predictions and human judgments across demographic groups on the Webpage Likability Prediction task.** Stronger correlations are observed in younger demographic groups, while performance gradually declines in older groups, likely reflecting limited training data used from older populations for the models. Pearson correlation coefficients ($r$) are reported between human demographic group ratings and Social Agents' simulated demographic group predictions, shown for three models: LLaMA 3.2 90B, Qwen2.5 VL 72B, and GPT-4o.

GPT-4o most closely matches human preferences, achieving $r = 0.71$ for 18-24 male and $r = 0.69$ for 18-24 female, both indicating strong linear agreement between Social Agents' predictions and the average human ratings from those demographic groups. Correlations decline steadily with age, with 55+ personas yielding $r$ between $0.22$ and $0.25$, indicating only weak linear alignment with the human ratings from the corresponding demographic groups. Across all models, female personas show consistently higher alignment, averaging $0.05$-$0.10$ higher Pearson correlations than male personas within the same age bracket. Similar trends are observed across the other models.

As noted in prior work on demographic performance disparities in LLMs (Liu et al., 2024), a likely explanation for the weaker correlations among older demographics is the training data composition. Most LLMs are trained predominantly on large-scale web and social-media corpora, where the available content reflects who is active on those platforms in the first place (Dodge et al., 2021). Participation in these platforms is itself strongly age-skewed: Pew Research Center data (Center, 2024) show that only 19% of U.S. adults aged 65+ use Instagram and just 10% use TikTok, compared with 76% and 59% of adults aged 18-29, respectively. Because older users post less of the content that ends up in scraped web corpora, their voices are correspondingly underrepresented in the data the model is trained on, and the latent representations learned by models reflect the aesthetic norms and preferences of younger cohorts. Older users, whose visual tastes are shaped by different cultural influences, design eras, and usage contexts, are therefore harder to simulate, leading to lower alignment with human judgments in those age groups.

### A.3.3 IMPACT OF NUMBER OF PERSONAS AND CALLS ON PREDICTIVE PERFORMANCE

We study how predictive performance varies with the computational budget $N$, defined as the total number of model calls allocated per item (i.e., per webpage or per ad) in Figure 6a and 6b. Budget can be spent in two ways: by repeatedly calling the same model prompt (increasing the number of calls), or by expanding the diversity of simulated persona agents. This is done by increasing the number of unique personas across demographic dimensions. We generate persona agents by sampling from a broad set of demographic and psychographic dimensions designed to capture a wide spectrum of human experiences and viewpoints. These dimensions include:

- **Age:** [18-29, 30-49, 50-64, 65+]
- **Education Level:** [Less than high school, High school graduate, Some college (no degree), Associate's degree, College graduate/some postgraduate, Postgraduate]
- **Gender:** [Male, Female]
- **Race/Ethnicity:** [White, Black, Asian, Hispanic]
- **Annual Income:** [$< \$30,000$, \$30K–\$100K, $> \$100K$]
- **Political Ideology:** [Liberal, Moderate, Conservative]
- **Political Affiliation:** [Democrat, Republican]
- **Religion:** [Protestant, Jewish, Atheist, Muslim, Hindu]

Let $D_i$ be the number of categories in demographic dimension $i$. The total potential persona space is:

$$|\mathcal{P}| = \prod_{i=1}^{8} D_i = 4 \times 6 \times 2 \times 4 \times 3 \times 3 \times 2 \times 5 = 17{,}280.$$

From this universe of 17,280 possible combinations, we curate and expand a subset of up to $N$ personas for our experiments.

To study how different allocations of this computational budget affect predictive performance, we compare the following five strategies:

- **No-Persona (Mean over $N$ Calls):** A No-Persona baseline with GPT-4o, where budget $N$ is spent on $N$ repeated calls using the same input prompt per webpage or ad.
- **Social Agents (10 Personas, $N/10$ Calls per Persona):** A fixed, panel of 10 persona agents, where given budget $N$, the calls per webpage or ad are evenly distributed across the panel such that each persona is queried $N/10$ times. For example, if $N = 20$, each persona is queried twice ($10 \times 2 = 20$ total calls).
- **Social Agents ($N$ Personas):** Here, the number and diversity of persona agents scale directly with the budget. For $N \leq 100$, we instantiate $N$ unique personas and query each once. Starting from a base pool $\mathcal{P}_{\text{base}}$ of 10 personas (five age groups $\times$ two genders), we define an expansion function

$$f : \mathcal{P}_{\text{base}} \times A \to \mathcal{P}$$

  where $A$ denotes a set of auxiliary demographic and trait attributes. By progressively augmenting $\mathcal{P}_{\text{base}}$ with attributes from $A$, we construct a larger persona pool containing up to 100 unique personas. For a given budget $N$, we sample $N$ distinct personas (e.g., if $N = 20$, 20 unique persona prompts are generated and each is used once).
- **Wisdom of the Silicon Crowd (WoSC) (Schoenegger et al., 2024):** $K$ **LLMs** $\times$ $(N/K)$ **Calls per LLM.** An ensemble of $K = 3$ distinct LLMs, $\mathcal{M} = \{\text{GPT-4o}, \text{Qwen3 32B}, \text{LLaMA 3.3 70B}\}$. Each model receives $\lfloor N/K \rfloor$ calls. For example, with $N = 20$, each model is queried 6 times, yielding $\approx N$ total calls. This setup tests the value of model diversity while keeping the total budget fixed.
- **WoSC $\times$ Social Crowd:** $K$ **LLMs** $\times$ $(N/K)$ **Personas.** A hybrid approach that combines model diversity with persona agent diversity. With $K = 3$ models and total budget $N$, each model $m_i$ is paired with $N/K$ distinct persona agents, queried once each. This ensures exactly $N$ persona agent-model calls per ad or webpage.

**Ad CTR Prediction Performance**

As shown in Figure 6 (a), the clearest gains emerge when scaling persona diversity. **Social Agents ($N$ Personas)** variant consistently delivers the lowest mean absolute percentage error (MAPE), dropping from about 32% at $N = 10$ to below 29% at $N = 100$. This steady decline highlights the value of incorporating an increasingly diverse set of personas, which broadens the range of simulated viewpoints and reduces systematic error. By contrast, the **WoSC** ensemble achieves a substantial reduction relative to the No-Persona baseline (mid 50% versus $\approx 71\%$ MAPE), but exhibits little to no further improvement as $N$ increases, demonstrating that model diversity alone plateaus quickly.

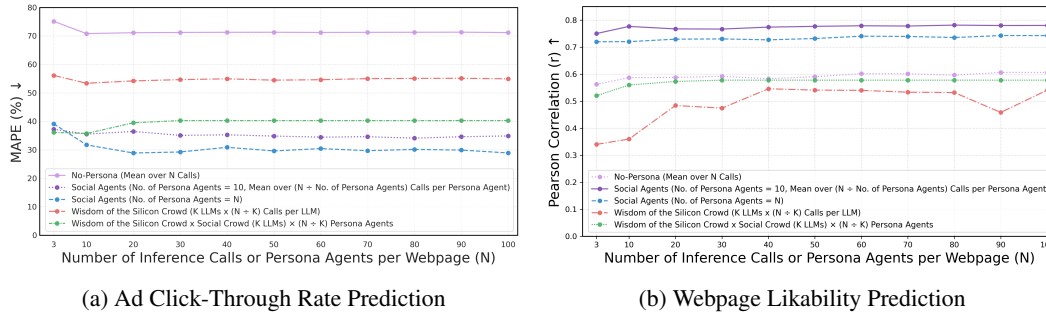

(a) Ad Click-Through Rate Prediction

(b) Webpage Likability Prediction

Figure 6: **Impact of number of inference calls or persona agents ($N$) on two tasks.** The $x$-axis denotes the per-item budget, realized either as $N$ independent calls to a single prompt or as $N$ persona agents with one call each for a given ad or webpage. **(a) Ad Click-Through Rate Prediction.** Evaluated using Mean Absolute Percentage Error (MAPE; lower is better). Performance plateaus by $N \approx$ 20-30, indicating that increasing the budget beyond this yields negligible gains. The best MAPE is achieved by *Social Agents* with $N = 20$ persona agents (also matched at $N = 100$), using one call per persona. **(b) Webpage Likability Prediction.** Evaluated using Pearson correlation ($r$; higher is better). Performance plateaus around $N \in \{20, 30, 40\}$, and the highest correlation is obtained by *Social Agents* with $N = 10$ persona agents and one call per persona.

The hybrid **WoSC** $\times$ **Social Crowd** settles near 39-40% MAPE, showing minimal benefit from larger budgets, as splitting calls across multiple models dilutes the effective persona coverage and constrains overall gains.

**Webpage Likability Prediction Performance**

Figure 6 (b) show that **Social Agents** with 10 Personas achieve the strongest Pearson correlations of $r \approx$ 0.74-0.76 across all budgets. This demonstrates that a carefully curated, representative panel is especially effective for subjective, aesthetic judgments. The **Social Agents ($N$ Personas)** variant also improves over both the **No-Persona** baseline ($r \approx$ 0.56) and the **WoSC** ensemble ($r \approx$ 0.35-0.55), but does not surpass the fixed 10 persona panel, suggesting that indiscriminately expanding persona agents may introduce noise rather than strengthen signal. The **WoSC** ensemble underperforms sharply, with unstable correlations that reflect the misalignment of some constituent models with this subjective task. The hybrid **WoSC** $\times$ **Social Crowd** provides modest improvements ($r \approx$ 0.56-0.59) over WoSC alone, yet remains below both Social Agents configurations.

### A.3.4 DIVERSITY ACROSS INTER-PERSONA PREDICTIONS IN SOCIAL AGENTS

To understand how simulated personas differ in their individual perspectives, we measure inter-persona prediction divergence on the Webpage Likability Prediction task using Wasserstein distance. Figure 7 presents a heatmap of Wasserstein distances between predicted likability distributions for ten persona archetypes defined by a age group combine with a gender. Each cell quantifies the distributional difference between two demographic groups' predictions, with higher values indicating greater divergence.

We observe the following patterns:

**Age-Based Divergence.** Age is the dominant source of divergence. Younger personas (18-24) show only moderate differences with adjacent groups (divergences of 0.26-0.39 against 25-34), but diverge strongly from older cohorts. For instance, the 18-24 female persona reaches distances of 0.58-0.83 when compared with 45+ groups, reflecting clear generational gaps in aesthetic preferences.

**Gender Effects.** Within the same age group, male-female divergences are present but smaller, typically ranging from 0.09 to 0.34. This indicates that gender contributes to variation in predictions.

**Extreme Divergence Cases.** The most pronounced divergence (0.83) occurs between 18-24 females and 45-54 males, reflecting maximal separation across both age and gender. The 45-54 male persona also shows consistently high divergence from younger groups (0.70-0.83). By contrast, older

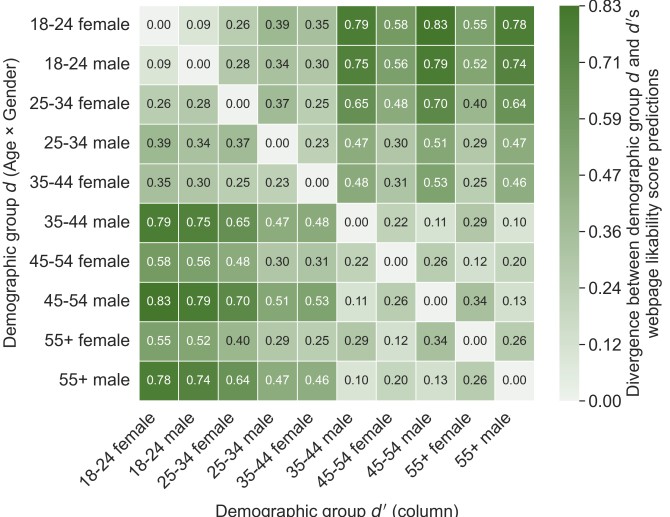

Figure 7: **Inter-Persona Prediction Divergence on the Webpage Likability Prediction task.** The heatmap shows clear clustering by age, with adjacent age groups exhibiting lower divergence (lighter cells along the diagonal) and sharp separations between younger females and older male groups (darkest cells, up to 0.83). Divergences between male and female personas within the same age bracket are comparatively smaller. Each cell reports the pairwise Wasserstein distance between predicted webpage-likability prediction distributions for GPT-4o when prompted with personas from different age and gender groups, where larger values indicate greater divergence in predictions.

males (55+) display slightly lower but still substantial divergence (0.64-0.78) from younger females, remaining significantly higher than within age comparisons.

### A.4 EFFECT OF AGGREGATION METHOD IN SOCIAL AGENTS

James Surowiecki's account of the Wisdom of Crowds identifies four core ingredients: diversity, independence, decentralization, and a mechanism for aggregation. While the literature often defaults to the median for its robustness to outliers and skewed responses, Hooker (Hooker, 1907), in a short commentary on Galton's ox-weight experiment, observed that the mean can be equally reliable, and sometimes even superior, when the underlying sample is structured and orderly rather than a random and noisy crowd. Galton himself (Galton, 1907) reported the median, but Hooker's observation suggests that mean aggregation is not inherently weaker; its effectiveness simply depends on the nature of the sample.

Our Social Agents framework more closely resembles this structured case. Personas are not arbitrary guesses but controlled social perspectives with systematic variation. Because the prediction set is organized rather than noisy, we treat aggregation as a tunable hyperparameter and directly compare mean and median across tasks. In every evaluation, the mean performs better, although both aggregation methods substantially outperform the No-Persona baseline.

As shown in Table 12, on the ad CTR prediction task, mean aggregation reduces GPT-4o error from 72.45 percent MAPE to 47.60 percent, while the median achieves 54.96 percent. On long-term video memorability prediction (Table 13), the mean reaches 0.41 Spearman correlation compared to 0.37 for the median and 0.33 for No-Persona. A similar pattern appears in webpage likability prediction (Table 14), where the mean attains 0.74 Pearson correlation and the median 0.71, both well above the 0.28 No-Persona baseline. Taken together, these results demonstrate that mean aggregation consistently yields stronger performance than median aggregation across all evaluated tasks, effectively capturing the collective signal across personas. At the same time, both mean and median aggregation consistently outperform single-agent baselines, underscoring the robustness of the Social Agents framework.

**Ad CTR Prediction (MAPE ↓)**

| Model | MAPE (%) |
|---|---|
| XGBoost (baseline) | 55.40 |
| LCBM (Zero-shot) | 80.21 |
| LCBM (Finetuned) | 75.67 |
| No-Persona (GPT-4o) | 72.45 |
| Social Agents (Mean) | 47.60 |
| Social Agents (Median) | 54.96 |

Table 12: Mean vs. median aggregation on Ad CTR prediction.

**Long-Term Video Memorability (Spearman Correlation ↑)**

| Model | Spearman Correlation |
|---|---|
| Human Consistency (baseline) | 0.55 |
| Henry (baseline) | 0.52 |
| No-Persona (GPT-4o) | 0.33 |
| Social Agents (Mean) | 0.41 |
| Social Agents (Median) | 0.37 |

Table 13: Mean vs. median aggregation on long-term memorability prediction.

### A.5 EFFECT OF PERSONA DISTRIBUTION AND DEMOGRAPHIC GROUNDING IN SOCIAL AGENTS

The effectiveness of Social Agents depends on how persona diversity is structured and whether demographic grounding is necessary for performance gains. We therefore analyze the role of persona distribution, scaling behavior, and demographic conditioning. We design personas by grounding each agent in demographic and psychographic dimensions commonly used in the behavioral and social sciences, including age, gender, education level, race and ethnicity, political ideology, political affiliation, income level, and religious background. In our main experiments, we use ten personas varied primarily by age and gender. We also conduct extended ablations in which personas are sampled across a broader multidimensional space.

The full space spans eight factors: age (18-29, 30-49, 50-64, 65+), education (less than high school through postgraduate), gender (male, female), race and ethnicity (White, Black, Asian, Hispanic), income (less than 30K, 30K-100K, more than 100K), political ideology (liberal, moderate, conservative), political affiliation (Democrat, Republican), and religion (Protestant, Jewish, atheist, Muslim, Hindu). This yields a theoretical universe of 17,280 persona combinations. From this universe, we curate subsets for experimentation. Scaling analyses show that performance improves as the number of agents increases from three to ten to twenty, but stabilizes beyond that point. Once major clusters of social viewpoints are covered, additional personas tend to introduce overlapping perspectives rather than new signal, and the marginal gains plateau.

The necessity of matching real-world demographic proportions depends on the availability of demographic-conditioned labels. Most benchmark datasets provide only aggregate labels without demographic annotations. In such low-resource settings, grounding to empirical population proportions is not feasible because there are no demographic-conditioned labels to align with. The objective is therefore not to mirror population frequencies, but to ensure coverage of diverse viewpoints. This is consistent with the Wisdom of Crowds principle, where diversity of opinion is the key requirement, while independence, decentralization, and aggregation are already enforced in our framework.

In high-resource settings where demographic labels are available, such as the webpage likability prediction task, we explicitly evaluate demographic-grounded persona conditioning. For each persona, we provide 5-shot examples whose labels correspond to the specific demographic group represented by that agent. This results in a modest but consistent improvement, where the standard Social Agents configuration achieves a Pearson correlation of 0.74 and 80.61 percent accuracy, while demographic-conditioned grounding improves performance to 0.76 correlation and 86.31 percent accuracy. The

| Webpage Likability Prediction (Pearson Correlation ↑) | |
| --- | --- |
| Model | Pearson Correlation |
| XGBoost (baseline) | 0.67 |
| No-Persona (GPT-4o) | 0.28 |
| Social Agents (Mean) | 0.74 |
| Social Agents (Median) | 0.71 |

Table 14: Mean vs. median aggregation on webpage likability prediction.

gains are incremental rather than dramatic, indicating that broad structured diversity captures most of the performance benefit, with fine-grained demographic grounding providing additional improvements when supported by labeled data.

Overall, our findings indicate that structured persona diversity alone is sufficient to activate the Wisdom of Crowds effect in low-resource conditions, while demographic grounding can further enhance performance when individual-level demographic labels are available.

## A.6   BIMODALITY IN BASELINE EVALUATIONS

Human ratings in our datasets exhibit the expected unimodal, approximately normal distributions characteristic of aggregated judgments. In contrast, model outputs under the No-Persona baseline display a markedly different pattern (Table 15).

For webpage likability prediction task, the No-Persona baseline shows clear bimodality. We extend this analysis to additional tasks and models under the same experimental setup. Using the mean of ten independent trials for GPT-4o, where the identical prompt is evaluated ten times and averaged, we observe bimodal prediction distributions across all three tasks: ad CTR prediction, long-term video memorability prediction, and webpage likability prediction. This suggests that an unconditioned model does not sample from a smooth continuum of viewpoints but instead oscillates between two dominant response modes.

| Model | Ad CTR Prediction | Long-Term Video Memorability | Webpage Likability |
| --- | --- | --- | --- |
| GPT-4o (No-Persona) | Bimodal | Bimodal | Bimodal |
| Qwen2.5 VL 72B (No-Persona) | Unimodal | Bimodal or Multimodal | Bimodal |

Table 15: Distributional structure of No-Persona baseline predictions across tasks.

For Qwen2.5 VL 72B, long-term video memorability displays bimodal or multimodal structure, and webpage likability is also bimodal. The only exception is ad CTR prediction, where Qwen produces a unimodal distribution.

Overall, while human judgments form smooth unimodal distributions, No-Persona LLM outputs frequently cluster into two peaks across tasks and models, highlighting a structural difference between human crowd judgments and neutral single-agent model sampling.

## A.7   CROWDS WITHIN VS. SOCIAL AGENTS

The concept of "crowds within" suggests that repeated estimates from a single individual can approximate a Wisdom of Crowds effect (Vul & Pashler, 2008). Prior behavioral research demonstrates that when an individual is prompted to reconsider an estimate under a different framing or assumption, the resulting second estimate draws on partly different knowledge and can partially replicate the benefits of crowd aggregation (Herzog & Hertwig, 2009). Recent work on LLMs reports similar findings, showing that sampling diverse reasoning paths and aggregating via majority vote also produces modest gains (Wang et al., 2023).

To examine this mechanism in our setup, we implement a crowds-within condition using two personas and induce additional variability through different decoding temperatures. Although this approach improves over a single No-Persona call for Ad-CTR prediction, it degrades performance on

the Memorability task. Moreover, the gains are substantially smaller than those achieved through the structured persona diversity of Social Agents (Table 16).

On long-term video memorability prediction, GPT-4o reaches a Spearman correlation of 0.20 under the crowds-within setup, compared to 0.41 with Social Agents using ten personas. Qwen3-32B attains 0.12 under crowds within, compared to 0.33 with Social Agents. A similar pattern appears in ad CTR prediction: GPT-4o achieves 56.65 percent MAPE under crowds within, compared to 47.60 percent with Social Agents; Qwen3-32B reaches 46.52 percent versus 43.31 percent under Social Agents.

| Long-Term Video Memorability | |
| --- | --- |
| Model | Spearman Correlation |
| Human Consistency | 0.55 |
| Henry (baseline) | 0.52 |
| No-Persona (GPT-4o) | 0.33 |
| Social Agents (GPT-4o) | 0.41 |
| No-Persona (Qwen3-32B) | 0.33 |
| Social Agents (Qwen3-32B) | 0.33 |
| GPT-4o (Crowds Within) | 0.20 |
| Qwen3-32B (Crowds Within) | 0.12 |

| Ad CTR Prediction | |
| --- | --- |
| Model | MAPE (%) |
| XGBoost | 55.40 |
| LCBM (Zero-shot) | 80.21 |
| LCBM (Finetuned) | 75.67 |
| No-Persona (GPT-4o) | 72.45 |
| Social Agents (GPT-4o) | 47.60 |
| No-Persona (Qwen3-32B) | 59.13 |
| Social Agents (Qwen3-32B) | 43.31 |
| GPT-4o (Crowds Within) | 56.65 |
| Qwen3-32B (Crowds Within) | 46.52 |

Table 16: Comparison of Crowds Within and Social Agents across tasks.

These findings suggest that crowds-within approaches rely primarily on stochastic variation around a shared underlying viewpoint. In contrast, Social Agents introduces structured heterogeneity across distinct demographic and psychographic perspectives. This form of diversity more closely aligns with classical Wisdom of Crowds conditions. As shown in our scaling experiments, once a sufficient number of distinct viewpoints is introduced, performance gains plateau, indicating that the latent diversity of the model has been effectively covered.

## A.8 Robustness Across Temperature Settings

We evaluate Social Agents across a range of decoding temperatures on one representative high-construal task, long-term video memorability prediction, and one representative low-construal task, ad CTR prediction.

On the long-term video memorability prediction (Table 17), for GPT-4o, Spearman correlations vary only slightly, from 0.40 at temperature 0.3, to 0.40 at 0.5, to 0.41 at 0.85, and 0.41 at 0.9. Qwen3-32B exhibits a similarly narrow range, between 0.32 and 0.35. As shown in Table 17, Social Agents consistently outperform the No-Persona baseline at every temperature, and the results remain tightly clustered.

| Long-Term Video Memorability (Spearman $\rho$) | | |
| --- | --- | --- |
| Temperature | GPT-4o | Qwen3-32B |
| 0.3 | 0.40 | 0.33 |
| 0.5 | 0.40 | 0.32 |
| 0.85 | 0.41 | 0.33 |
| 0.9 | 0.41 | 0.35 |
| Baseline | 0.33 | 0.33 |

Table 17: Social Agents performance across temperatures on long-term video memorability prediction.

We observe the same pattern on the ad CTR prediction task. GPT-4o remains clustered around 47.5 percent MAPE across temperatures, while Qwen3-32B varies within a narrow band. In all cases, Social Agents outperform the No-Persona baselines.

| Ad CTR Prediction (MAPE $\downarrow$) | | |
|---|---|---|
| Temperature | GPT-4o | Qwen3-32B |
| 0.3 | 47.71 | 45.67 |
| 0.5 | 47.53 | 44.86 |
| 0.85 | 47.60 | 43.31 |
| 0.9 | 47.57 | 46.65 |
| Baseline | 72.45 | 59.13 |

Table 18: Social Agents performance across temperatures on low-construal CTR prediction.

Taken together, these findings demonstrate that Social Agents are robust to temperature-based variability across both construal levels. Adjusting the sampling temperature produces only minor fluctuations and does not alter the underlying decision structure. The gains of Social Agents arise from structured persona-level diversity rather than stochastic variation introduced through decoding. While higher temperatures may allow slightly greater creativity and occasional marginal gains, overall behavior remains stable across models and tasks.

