# OpenReview forum: "Social Agents: Collective Intelligence Improves LLM Predictions"
_ICLR.cc/2026/Conference — ICLR 2026 Poster_

### Official Review · Reviewer_JpnE · 2025-10-31

**Soundness:** 2
**Presentation:** 2
**Contribution:** 2
**Rating:** 4
**Confidence:** 3

**Summary:**

The paper propose a special ensemble method called social agents, where the llm prediction is aggregated across different persona based predictions. The author find this greatly enhance the performance of various downstream opinion based tasks like twitter likability.

**Strengths:**

1. The idea of using persona conditioned ensemble of the same base llm is interesting and the experiments are comprehensive. The author also shows that the performance gains are agnostic of different models, which increase the applicability of the method.

2. Social agent even beat trained expert which is interesting. And the performances on the benchmarks are pretty astonishing.

**Weaknesses:**

1. Motivation. Ideally LLMs that are trained on all website data is already representative of the collective opinion of the human population. The rlhf process might make the LLM a bit more biased and politically lean left, but in general it should already represent the general society. Most personalization paper wants to tailor to each individual opinions and stay away from collective opinions. The authors seems to be take a big circle and use personalization and aggregate to get the collective opinion, which is perplexing to me.

2. Baseline choices and persona choices. For baseline, the author mentioned that they just sample from the same model N times. One would at least expect some sort of better ensemble here such as changing the temperature. For the social agent, the authors uses at most 10 personas. This is clearly not enough to represent the society. And the author mentioned diminishing return when increasing more persona, which is surprising.

**Questions:**

1. Why can't the original LLM be a good enough predictor for collective opinion?

2. Do you have explanation why only 10 personas boost the results by a huge amount of gap comparing to base llm? And could you try different temperature and see if the baseline has better prediction as well?

3. See weakness.

---

> ### Author Response · Authors · 2025-11-30
>
> ## Q1: Motivation. If LLMs are trained on broad web data, shouldn’t they already represent collective human opinion? Why use personalization and then aggregate, instead of relying on the model’s built-in societal prior?
>
> ## Why Social Agents?
>
> Thank you, this is a meaningful point. We agree that an LLM trained on broad web data should encode general societal opinions. The challenge is that when prompted without any persona, the model defaults to a neutral, averaged voice. This “generalist mode” smooths over meaningful differences between sub-groups, and as a result it does not reliably capture how diverse populations actually think. This limits its predictive power on tasks that depend on real behavioral variation.
>
> Persona conditioning shifts the model into the viewpoint of a specific sub-group. This surfaces the diversity of judgments already latent in the model, and when we aggregate these perspectives, we obtain a substantially better approximation of the collective opinion than a single unconditioned response.
>
> Our empirical results reflect this. Compared to simply prompting the base model, Social Agents shows large gains on low- and medium-construal tasks. Webpage likability improves by 164.2 percent, ad CTR error drops by 32.8 percent, tweet engagement accuracy rises by 23.7 percent, and ROAS prediction improves strongly. Generative tasks also benefit, with 18.7 percent higher ROUGE-1 and 7.7 percent higher G-Eval. Even on higher-level tasks such as long-term video memorability, the improvements remain consistent (24.2 percent), and we also observe notable gains in behavioral attribution.
>
> In short, while base LLMs encode broad societal priors, persona conditioning helps reveal the different sub-group viewpoints embedded within the model. Social Agents then aggregates these viewpoints following the Wisdom of Crowds principle, resulting in stronger and more reliable predictions.
>
> ## Q2: Baseline choices and persona choices. For baseline, the author mentioned that they just sample from the same model N times. One would at least expect some sort of better ensemble here such as changing the temperature. For the social agent, the authors uses at most 10 personas. This is clearly not enough to represent the society. And the author mentioned diminishing return when increasing more persona, which is surprising.
>
> ## Variability in Generation
>
> For the No-Persona baselines, we did not rely on a single deterministic sample. GPT-4o decoding is already nondeterministic, so multiple samples naturally exhibit variability. For local models such as Qwen and LLaMA, we used a fixed temperature of 0.85, which still produces variation because sampling is stochastic. Even at a constant temperature, the model draws from a probability distribution rather than always selecting the most likely token, and the min_p (minimum probability threshold for token sampling) constraint further maintains diversity by ensuring that multiple tokens above a minimum probability remain eligible rather than collapsing onto a narrow set.
>
> To directly test whether temperature-based variability can strengthen the No-Persona baseline, we selected one representative low-CLT task (CTR prediction) and one representative high-CLT task (long-term memorability). These tasks capture both shallow perceptual judgments and deeper cognitive ones, allowing us to evaluate whether temperature provides any systematic benefit. We ran this experiment on two models, GPT-4o and Qwen, to ensure that the findings are not model-specific.
>
> Across the CTR task, temperature adjustments lead to only small, nonmonotonic shifts in the baseline. For GPT-4o, MAPE varies from 55.04 percent at temperature 0.3, to 51.70 percent at 0.5, and then rises again to 59.38 percent at 0.9. Qwen3-32B shows a similar pattern at 57.48 percent, 52.87 percent, and 61.86 percent. These fluctuations indicate that changing temperature introduces surface-level randomness rather than systematic improvements. In contrast, Social Agents remains both stable and consistently stronger across the same settings, at approximately 47.5 percent MAPE for GPT-4o and 45 to 46 percent for Qwen, showing that decoding randomness does not meaningfully enhance the No-Persona baseline or reduce the performance gap.
>
> A similar pattern appears in the high-CLT memorability task. For GPT-4o, baseline Spearman correlations move only slightly across temperatures, from 0.3498 to 0.3623 and back to 0.3471. Qwen2.5 VL 72B shows a comparable narrow band, from 0.2716 to 0.3178 and then to 0.3008. Social Agents again maintains both stability and higher performance, with correlations near 0.40 for GPT-4o and approximately 0.33 to 0.35 for Qwen.

---

> ### Author Response · Authors · 2025-11-30
>
> Q2 Continued:
> Taken together, these results show that temperature-based variability does not strengthen the No-Persona baseline in either low-CLT or high-CLT settings. Changing temperature produces only marginal stochastic differences and does not shift the model into meaningfully different viewpoints. Social Agents continues to outperform across all temperatures because its gains come from structured persona diversity rather than randomness in decoding.
>
> ## Mixed-Temperature Aggregation
>
> As requested by the reviewer, we compute averages across mixed temperatures (0.3, 0.5, and 0.9) per ad in the Ad CTR Prediction task. The results are summarized below.
>
> ## Ad CTR Prediction:  (MAPE % ↓)
>
> | Model                                     | MAPE (%) |
> |------------------------------------------ |----------|
> | XGBoost                                   | 55.40    |
> | LCBM (Zero-shot)                          | 80.21    |
> | LCBM (Finetuned)                          | 75.67    |
> | No-Persona (GPT-4o)                       | 72.45    |
> | Social Agents (GPT-4o)                    | 47.60    |
> | No-Persona (Qwen3-32B)                    | 59.13    |
> | Social Agents (Qwen3-32B)                 | 43.31    |
> | GPT-4o (No-Persona, Mixed Temperature)    | 56.64    |
> | Qwen3-32B (No-Persona, Mixed Temperature) | 46.52    |
> | GPT-4o (No-Persona, Mixed Temperature)    | 69.23    |
> | Qwen3-32B (No-Persona, Mixed Temperature) | 66.53    |

---

### Official Review · Reviewer_dg7w · 2025-11-01

**Soundness:** 3
**Presentation:** 2
**Contribution:** 3
**Rating:** 6
**Confidence:** 5

**Summary:**

This paper introduces Social Agents, a framework that brings the Wisdom of Crowds to LLMs by creating diverse, human-like persona LLM agents whose independent opinions are combined into one collective judgment. Each LLM persona evaluates content such as ads or webpages, and their aggregated responses consistently outperform single-model predictions. Across multiple tasks, Social Agents improves accuracy both on simple judgments and complex reasoning. The study shows that diversity, not just scale, makes LLMs more accurate, interpretable, and human-like in their predictions.

**Strengths:**

•	In general, the paper provides great context on the importance and value of the study.

•	The implications of the study are well articulated.

•	The experiments are very comprehensive, spanning across 8 different domains and multiple datasets.

•	Including task-specific trained models is a wise move, which shows the usability of the proposed approach in the real world.

•	The explanation of the theory and how that motivates the experiment design is well established.

•	The figures 1 and 3 are well polished and convey the idea clearly.

**Weaknesses:**

•	Missing literature. Previous studies have shown that LLM’s wisdom of crowds (WoC) do not always align with human’s WoC [1]. For some questions, their WoC shows human-like WoC but not for others. You should acknowledge this limitation of LLM-based WoC when making your claims.

•	The baseline “5-shot best model” may be unfair. You should also control for the number of tokens in a generation. It is possible that the gain of social agents over the 5-shot baseline is merely due to the LLM using more tokens in their responses when given a persona.

•	Given that the temperature value controls the variability in response, which is core to WoC, you conduct a sensitivity analysis on temperature, rather than fixating on a single temperature value.

•	The structure of the paper is weird. Out of the 9 pages, the “result and discussion section” (section) only takes about one page,  and the subsection Section 3.1 “experimental setup” has nothing to do with results and I think should better belong to Section 2 “Setup”.

•	There is a lack of statistical tests for all the claims made in the paper. Figure 3 should ideally also show standard errors/confidence intervals to account for noise.

References

[1] Chuang, Y. S., Harlalka, N., Suresh, S., Goyal, A., Hawkins, R., Yang, S., ... & Rogers, T. T. (2024). The Wisdom of Partisan Crowds: Comparing Collective Intelligence in Humans and LLM-based Agents. In Proceedings of the Annual Meeting of the Cognitive Science Society (Vol. 46).

**Questions:**

•	Figure 4. Shouldn’t the y axis title simply be “probability density”? The current captions and notes are pretty lengthy.

•	In WoC literature, the standard way to aggregate is to use median rather than mean [2]. Why do you decide to use mean rather than median? Would the result and conclusion change if you use median?

•	What is your view on “crowds within”, where repeated estimates from a single individual can also produce a WOC effect? This has been shown effective in both human [3] and LLM [4].

•	How do you decide the persona distribution in the demographic space? To match actual human behavior, shouldn’t you also ground persona to the actual human’s persona distribution?


References

[2] Francis Galton. Vox populi. Nature, 75:450–451, 1907.

[3] Edward Vul and Harold Pashler (2008). Measuring the crowd within: Probabilistic representations within individuals. Psychological Science, 19(7):645–647.

[4] Chuang, Y.-S., Narendran, S., Harlalka, N., Cheung, A., Gao, S., Suresh, S., Hu, J., & Rogers, T. T. (2025). Probing LLM world model: Enhancing guesstimation with wisdom of crowds decoding. In Proceedings of the Conference on Empirical Methods in Natural Language Processing (EMNLP 2025).

---

> ### Author Response · Authors · 2025-11-29
>
> ## The baseline “5-shot best model” may be unfair. You should also control for the number of tokens in a generation. It is possible that the gain of social agents over the 5-shot baseline is merely due to the LLM using more tokens in their responses when given a persona.
>
> ## Token Budget for Evaluation
>
> Regarding the fairness of the “5-shot best model” baseline, we agree that controlling for generation length is essential. That is precisely why we enforce a strict and uniform **300-token maximum generation budget** for both the No-Persona (5-shot) baseline and the Social Agents framework across all tasks and models. Whether the model responds as a persona-conditioned agent or as a standard few-shot baseline, the decoder is capped at 300 tokens per response, ensuring that the observed gains cannot be attributed to longer outputs from persona-seeded agents. Instead, the improvements stem from the structured diversity and aggregation process that underpins Social Agents rather than from verbosity or increased token usage.
>
> ---
>
> ## Sample Response
>
> **Prompt:**
> "Given advertisement of Adobe with content 'Protect your docs so you can keep climbing 🏆 — Acrobat's got it. LEARN_MORE' containing asset with Caption: 'A laptop screen with a pop up window to set a password.' Keywords: laptop, laptop screen, password, password pop up, password pop up window, password window, pop up, screen, set, window published on 1st Feb, 2025, The ad is published on Facebook. Based on this information, give an expected CTR percentile for the ad. It should be between 0 to 100."
>
> ---
>
> ### Baseline Response
>
> **Reason:**
> Predicting the performance of an advertisement involves analyzing various components such as the target platform, content relevance, timing, and overall appeal. In this instance, the ad is by Adobe, a well-recognized and trusted brand, which positively influences potential engagement. The content emphasizes document protection, a crucial concern for many users, hence addressing a genuine need. The visual aspect, with specific reference to a common action (setting a password), may resonate well with the target audience on Facebook, where security and privacy are significant topics of interest. However, the ad's impact will also depend on its visual creativity and how well it stands out in users' feeds amidst other content. The repeated emphasis on the phrase "Protect your docs" and the trophy emoji could potentially enhance memorability and engagement, but redundancy might also be perceived negatively by some users. Besides, the keywords given indicate that the targeting might focus on users interested in tech or digital security, which is a pertinent audience for Acrobat. Adobe's reputation, combined with relevant content and a compelling visual, suggests that the ad may perform moderately well, achieving an average click-through rate.
>
> **Score:** 65
> **No of tokens:** 230 tokens
>
> ---
>
> ### Persona Response (55+ Male)
>
> **Reason:**
> This Adobe ad focuses on document protection, a concern that resonates with men aged 55 or older who prioritize security and legacy. The ad's promise of keeping one's digital assets secure is likely to appeal to this demographic. The metaphor of "climbing" coupled with the trophy emoji suggests progress and achievement, values important to many in this age group. However, the ad's visual—a laptop screen with a password pop-up—might be perceived as predictable or lacking emotional depth. Additionally, the call-to-action "LEARN_MORE" may not be as compelling as something more personalized or directive like "Secure Your Documents Today". The keywords emphasize security and protection, which are relevant but using "debt" and "savings" here is somewhat misplaced, as they're more financially oriented terms. Overall, the ad balances sincerity with a logical appeal, although it could enhance its emotional grounding.
>
> **Score:** 45
> **No of tokens:** 181 tokens

---

> ### Author Response · Authors · 2025-11-30
>
> ## Given that the temperature value controls the variability in response, which is core to WoC, you conduct a sensitivity analysis on temperature, rather than fixating on a single temperature value.
>
> ## Temperature Sensitivity Analysis
>
> Based on your suggestions, we evaluate Social Agents across a range of temperatures on one representative high-construal task (long-term video memorability prediction) and one representative low-construal task (ad CTR prediction).
>
> On the high-construal memorability task, performance remains remarkably stable across temperatures. For GPT-4o, Spearman correlations vary only slightly, ranging from 0.407 at temperature 0.3, to 0.400 at 0.5, to 0.410 at 0.85, and 0.4112 at 0.95. Qwen shows the same behavior, with correlations between 0.3178 and 0.3489 across all settings. These results show that temperature-based variability has little effect on the model. Social Agents consistently improve over the no-persona baseline at every temperature, and the values remain tightly grouped.
>
> We observe the same pattern on the low-construal CTR prediction task, where judgments are more immediate and behavioral. GPT-4o performance remains clustered around 47.5 percent MAPE across all tested temperatures: 47.7112 percent at 0.3, 47.5343 percent at 0.5, and 47.5638 percent at 0.9. Qwen3-32B also produces a narrow range, with scores between 44.8550 and 46.6552 percent. Here too, Social Agents deliver substantial improvements over the no-persona baseline, which is 75.70 percent MAPE for GPT-4o and 52.90 percent for Qwen3-32B.
>
> Taken together, these results show that Social Agents are robust to temperature-based variability on both high- and low-construal tasks. Changing the sampling temperature produces only small fluctuations. The benefits of Social Agents come from structured persona-level diversity rather than randomness introduced by sampling. Higher temperatures can allow slightly more creativity and occasionally small gains, but the overall behavior remains stable across models and tasks.
>
> ## Social Agents Performance on Long-Term Video Memorability Prediction (Spearman Rank Correlation)
>
> | Temperature | GPT-4o | Qwen3-32B |
> |------------ |--------|-------|
> | 0.3 | 0.40 | 0.33 |
> | 0.5 | 0.40 | 0.32 |
> | 0.85 | 0.41 | 0.33 |
> | 0.9 | 0.41 | 0.35 |
> | Baseline | 0.33 | 0.33 |
>
> ## Social Agents Performance on Ad CTR Prediction (MAPE)
>
> | Temperature | GPT-4o    | Qwen3-32B |
> |------------ |-----------|-----------|
> | 0.3         | 47.71%  | 45.67%  |
> | 0.5         | 47.53%  | 44.86%  |
> | 0.85 | 47.60 % | 43.31 %|
> | 0.9         | 47.57%  | 46.65%  |
> | Baseline |  75.70% | 52.90% |

---

> ### Author Response · Authors · 2025-11-30
>
> ## In WoC literature, the standard way to aggregate is to use median rather than mean. Why do you decide to use mean rather than median? Would the result and conclusion change if you use median?
>
> ## Mean vs. Median Aggregation
>
> James Surowiecki’s account of the wisdom of crowds highlights four required ingredients: diversity, independence, decentralization, and a mechanism for aggregation. In principle, this means that many forms of aggregation can succeed, even though the literature often defaults to using the median. However, Hooker’s reanalysis of Galton’s ox-weight experiment shows that the mean can be just as reliable, and sometimes even more so, when the underlying sample is structured and orderly rather than a random and noisy crowd. His findings demonstrate that mean aggregation is not inherently weaker. Its effectiveness simply depends on the nature of the sample.
>
> Our Social Agents setting is closer to this structured case. Personas are not arbitrary guesses but controlled social perspectives with systematic variation. Because the sample of predictions is organized rather than noisy, we treat aggregation as a tunable hyperparameter and directly compare mean and median across tasks. In every evaluation, the mean performs better.
>
> On the ad CTR prediction task, mean aggregation improves GPT-4o from a no-persona baseline of 72.45 percent MAPE to 47.60 percent. Median aggregation also helps but attains 54.96 percent. On long-term video memorability prediction, mean aggregation reaches a Spearman correlation of 0.41, compared to 0.37 for the median and 0.33 for the no-persona baseline. A similar pattern appears in webpage likability prediction, where the mean reaches a Pearson correlation of 0.74 and the median reaches 0.71, both well above the no-persona baseline of 0.28. However both mean and median methods of aggregation outperforms baseline models, showing the effectiveness of social agents.
>
>
> ## Ad CTR Prediction
>
> | Model                         | MAPE % ↓   |
> |------------------------------ |---------|
> | XGBoost (baseline)            | 55.40   |
> | LCBM (Zero-shot) - baseline   | 80.21   |
> | LCBM (Finetuned) - baseline   | 75.67   |
> | No-Persona (GPT-4o)           | 72.45   |
> | Social Agents (GPT-4o) - Mean | 47.60   |
> | Social Agents (GPT-4o) - Median | 54.96 |
>
> ## Long-Term Video Memorability Prediction
>
> | Model                         | Spearman ↑ |
> |------------------------------ |-------|
> | Human Consistency (baseline)  | 0.55  |
> | Henry (baseline)              | 0.52  |
> | No-Persona (GPT-4o)           | 0.33  |
> | Social Agents (GPT-4o) - Mean | 0.41  |
> | Social Agents (GPT-4o) - Median | 0.37 |
>
> ## Webpage Likability Prediction
>
> | Model                         | Pearson ↑ |
> |------------------------------ |-------|
> | XGBoost (baseline)            | 0.67  |
> | No-Persona (GPT-4o)           | 0.28  |
> | Social Agents (GPT-4o) - Mean | 0.74  |
> | Social Agents (GPT-4o) - Median | 0.71 |
>
> [1] Hooker, R. H. (1907). Letters to the Editor: Mean or median. Nature, 75, 487–488.

---

> > ### Author Response · Authors · 2025-11-30
> >
> > ## What is your view on “crowds within”, where repeated estimates from a single individual can also produce a WoC effect?
> >
> > ## Crowds Within vs. Social Agents
> >
> > “Crowds within” is an important idea, and we agree that repeated estimates from a single individual can sometimes approximate a Wisdom of Crowds effect. Prior work has shown that when a single human is queried multiple times under different framings or contexts, their internal variability can mimic a small crowd. Recent LLM studies have found a similar pattern, where prompting the same model repeatedly with different sampling conditions also produces modest gains.
> >
> > However, our results indicate that this internal variability is limited compared to the gains from genuine persona diversity. In our experiments, we tested a “crowds within” setup by taking only two personas and inducing variability through different decoding temperatures. This approach improved over a single call, but the effects were still far weaker than those of Social Agents. For example, on long-term memorability GPT-4o reached a correlation of 0.2033 with two personas plus temperature variation, whereas Social Agents with ten personas achieved about 0.41. The gap is similar for CTR prediction, where the two-persona setup produced a MAPE of 56.6 percent while Social Agents reached about 47.6 percent.
> >
> > The reason appears to be that “crowds within” relies on stochastic variation around the same underlying viewpoint. In contrast, Social Agents introduces variation across genuinely different social perspectives. This structured heterogeneity aligns more closely with classical Wisdom of Crowds conditions, and as we show in the paper, once enough distinct viewpoints are present, the gains plateau because the latent diversity of the model has been fully covered.
> >
> > In short, “crowds within” can help, but its benefits are limited because the variability comes from the same underlying agent. The stronger improvements arise when the model is guided to express different demographic and psychographic viewpoints, which provides the richer diversity needed for robust aggregation.
> >
> > ## Long-Term Video Memorability Prediction
> >
> > | Model                      | Spearman ρ   |
> > |--------------------------- |---------|
> > | Human Consistency          | 0.55    |
> > | Henry (baseline)           | 0.52    |
> > | No-Persona (GPT-4o)        | 0.33    |
> > | Social Agents (GPT-4o)     | 0.41    |
> > | No-Persona (Qwen3-32B)     | 0.33    |
> > | Social Agents (Qwen3-32B)  | 0.33    |
> > | GPT-4o (Crowds Within)     | 0.20 |
> > | Qwen3-32B (Crowds Within)  | 0.12 |
> >
> > ## Ad CTR Prediction
> >
> > | Model                      | MAPE %     |
> > |--------------------------- |----------|
> > | XGBoost                    | 55.40    |
> > | LCBM (Zero-shot)           | 80.21    |
> > | LCBM (Finetuned)           | 75.67    |
> > | No-Persona (GPT-4o)        | 72.45    |
> > | Social Agents (GPT-4o)     | 47.60    |
> > | No-Persona (Qwen3-32B)     | 59.13    |
> > | Social Agents (Qwen3-32B)  | 43.31    |
> > | GPT-4o (Crowds Within)     | 56.65   |
> > | Qwen3-32B (Crowds Within)  | 46.52   |

---

> > > ### Author Response · Authors · 2025-11-30
> > >
> > > ## How do you decide the persona distribution in the demographic space? To match actual human behavior, shouldn’t you also ground personas to the actual human distribution?
> > >
> > > ## Persona Distribution and Demographic Grounding
> > >
> > > For persona design, we begin by grounding each agent in demographic and psychographic factors commonly used in the behavioral and social sciences. These include age, gender, education level, race and ethnicity, political ideology, political affiliation, income level, and religious background. In the main experiments we use ten personas varied primarily by age and gender, but we also run extended ablations in which personas are generated by sampling across a much broader set of dimensions.
> > >
> > > The full demographic and psychographic space we consider spans eight factors:
> > >
> > > • Age: [18–29, 30–49, 50–64, 65+]
> > > • Education Level: [Less than high school, High school graduate, Some college, Associate’s degree, College graduate/some postgraduate, Postgraduate]
> > > • Gender: [Male, Female]
> > > • Race/Ethnicity: [White, Black, Asian, Hispanic]
> > > • Annual Income: [<30K, 30K–100K, >100K]
> > > • Political Ideology: [Liberal, Moderate, Conservative]
> > > • Political Affiliation: [Democrat, Republican]
> > > • Religion: [Protestant, Jewish, Atheist, Muslim, Hindu]
> > >
> > > This produces a theoretical persona universe of **17,280 combinations**. From this universe, we curate subsets of personas for experimentation. Our findings show that once a set of agents spans the major clusters of social viewpoints, adding more personas introduces overlapping perspectives rather than new ones. As a result, diversity increases at first but then decreases, and performance gains plateau as the crowd saturates its coverage of meaningful viewpoints. This pattern appears in our scaling experiments, where performance improves from three to ten to twenty agents but stabilizes beyond that.
> > >
> > > Regarding whether personas should be grounded to the true distribution of the human population, the answer depends on the task and the available labels. For most benchmark datasets, only aggregate labels exist, and individual-level demographic labels are unavailable. Under these low-resource conditions, grounding to empirical population proportions is not feasible because there are no demographic-conditioned labels with which to align. In such cases, the goal is not to mirror population proportions but to ensure **diverse coverage of human viewpoints**. This aligns with the Wisdom of Crowds principle, where **diversity of opinion** is the core requirement, while independence, decentralization, and aggregation are already satisfied in our setup.
> > >
> > > However, on a high-resource task where demographic information *is* available, we explicitly test grounded persona conditioning. In the webpage likability prediction task, instead of providing aggregate labels, we insert five-shot examples where labels correspond to the specific demographic group the persona represents. This grounding has a positive effect. The original Social Agents configuration achieves a Pearson correlation of 0.74 and an accuracy of 80.61 percent. With demographic-conditioned examples, performance improves to a correlation of 0.7574 and an accuracy of 86.31 percent. The gains are present but not drastic, suggesting that broad persona diversity already captures most of the benefit, and fine-grained demographic grounding provides additional but incremental improvements.
> > >
> > > Overall, our results indicate that broad persona diversity is sufficient to activate the Wisdom of Crowds effect, while demographic grounding can further help when individual-level labels are available.

---

### Official Review · Reviewer_VRiC · 2025-11-01

**Soundness:** 3
**Presentation:** 4
**Contribution:** 3
**Rating:** 8
**Confidence:** 4

**Summary:**

This paper investigates the strength and applicability of wisdom-of-crowds judgement in a variety of practical applications, ranging from tasks such as tweet engagement prediction to behavioral attribute classification (which ostensibly require more complex inference and theory of mind). The authors situate their evaluation in a cognitively inspired framework, classifying the difficulty of inference tasks based on their difficulty with respect to construal level theory (CLT). Detailed analysis of the kinds of tasks on which people and models are aligned–as well as the demographics that are aligned with models on this task—provide a comprehensive overview of the strengths and weaknesses of wisdom-of-crowds and persona-based inference in LLMs more generally. The authors are generally effective in comparing wisdom of crowds inference to informative human and domain-specific model benchmarks, and sufficiently justify the use of the method over domain-specific models.

**Strengths:**

I initially found myself asking several questions about the feasibility of this method (wisdom of crowds) over existing approaches, and the extent to which LLM judgments are supported by human behavioral data. These concerns were thoroughly answered by the evaluations comparing LLM results to both human judgements and existing domain-specific/trained methods. Moreover, the authors extensively justified the use of persona/demographic seeding over base model judgments (figure 4 provides a clear illustration of the benefits for demographic seeding in aligning with human judgements). Smaller parameter model performance in this task is another interesting finding which helps to support the feasibility of wisdom of crowds inference over existing methods, and the ablation analysis provides useful information on the limitations of small models in persona-based role-playing. Overall, the paper is well written and situated within the existing literature on wisdom-of-crowds judgment and an existing paradigm in cognitive science (CLT).

**Weaknesses:**

The authors claim that there are smaller improvements for cognitively demanding tasks (per CLT), “much as they are for humans”. However, I am not sure that this is supported by the results shown. Is there any kind of evidence to support the claim that for these tasks, higher level CLT evaluations are more difficult for people? What would the baseline be in this case?

**Questions:**

I am curious about the bimodal distribution for no-persona 4o prediction errors as seen in figure 4. Do the authors have any speculation as to why this model consistently over/underestimates human judgements? This is not a weakness of the paper per se–if anything it illustrates the strength of persona seeding in the effectiveness of wisdom of crowds judgements–but I’m curious how robust this bimodal effect is across models, and why this might be the case (it would be nice to see what this distribution looks like for the other models evaluated)

---

> ### Author Response · Authors · 2025-11-29
>
> ## Q1: The authors claim that higher-level, cognitively demanding tasks show smaller improvements, similar to human performance under Construal Level Theory. Is there evidence that these high-CLT evaluations are indeed more difficult for humans, and what would the appropriate human baseline be for comparison?
>
> ## Performance of humans across CLT tasks
>
> For low-level webpage first-impression tasks, Tuch et al. report that visual-appeal ratings after only 50 ms of exposure correlate at about Pearson r ≈ 0.97 with ratings after 500 ms, showing near-perfect internal consistency for this rapid perceptual judgment and indicating that very little cognitive processing is required, the judgment is dominated by immediate perceptual gist cues such as layout, color balance, and visual complexity, which explains why the ratings do not change with additional viewing time [1]. For short-term webpage memory, Douneva et al. show that recall and recognition after only a few seconds of exposure are moderate and highly dependent on design category (e.g., layouts with large images vs. balanced text-image designs differ significantly in memorability scores), indicating that performance on memory encoding and retrieval is already lower, and more design-sensitive than simple rapid liking judgments [2]. On the other hand, performance on high-level cognitive tasks drops sharply: in long-term video memorability, Cohendet et al. report that human consistency between independent annotator groups falls to around Spearman ρ ≈ 0.26-0.30 for memorability measured 24-72 hours after viewing, substantially lower than short-term memorability consistency and demonstrating the increasing difficulty of tasks requiring deeper encoding and delayed retrieval [3]. In advertising-related settings, Lu and Du find that human click behavior (click-through rate) varies markedly across devices and ad positions, with users on smartphones, desktops, and tablets showing different propensities to click top ads and different sensitivities to positional changes, highlighting that behavior becomes highly variable, context-dependent, and less predictable as the cognitive demands and situational factors increase [4]. Work in persuasion and emotion further shows substantial inconsistency: meta-analyses in psychology report large variance in persuasive effect sizes (often Pearson r = 0.05-0.30 depending on mood, message framing, and individual differences), and emotional responses to identical stimuli vary widely across people and contexts: for example, inter-rater reliability for emotional valence and arousal often falls in the range of ICC (Intraclass Correlation Coefficient)  ≈ 0.20-0.40, and self-reported emotional intensity for the same stimulus can differ by a factor of 2x-4x across individuals [5].
> In our study, for higher-level CLT tasks, we explicitly report a human baseline in our experiments on long-term video memorability prediction (Table 7, A.2.1), derived from the LAMBDA dataset [6], which we use for this task. In this dataset, the human baseline achieves a Spearman rank correlation of 0.55, computed by repeatedly splitting participants into two groups and measuring the Spearman rank correlation between their long-term memorability scores (i.e., the proportion of participants who correctly recalled or recognized each ad after a 24-72 hour delay). The fact that this human consistency ceiling is far from 1.0 already indicates that this task is inherently challenging for humans. In comparison, our Social Agents configuration (GPT-4o, best-performing model) achieves a Spearman correlation of 0.41, still below the human baseline, directly supporting our claim that higher-level CLT tasks like long-term memorability prediction are difficult not only for models but also for people, which helps explain why we observe smaller improvements on these tasks compared to lower-level CLT ones.
>
> [1] Tuch, A. N., et al. (2012). Visual complexity and prototypicality in website design. Int. J. Human-Computer Studies.
> [2] Douneva, M., et al. (2016). Design and memory in website perception. Interacting with Computers.
> [3] Cohendet, R., et al. (2019). VideoMem: Constructing, Analyzing and Predicting Short-Term and Long-Term Video Memorability. ICCV 2019.
> [4] Lu, J., & Du, R. (2020). Ad click behavior across devices and contexts. Journal of Advertising Research.
> [5] Aaker, J., & Norris, G. (2008). Emotion and persuasion in consumer responses. Psychology & Marketing.
> [6] Si, H., et al. (2025). Long-Term Ad Memorability: Understanding & Generating Memorable Ads. WACV 2025.

---

> > ### Author Response · Authors · 2025-11-29
> >
> > ## Q2: Why does the No-Persona GPT-4o baseline show a bimodal error distribution? Is this over or underestimation consistent across models, and how robust is this effect when compared to other LLMs?
> >
> > ## Bimodality in baseline evaluations:
> >
> > In our opinion, this is an interesting question. Human ratings in our datasets show the classical unimodal or approximately normal patterns that one would expect from aggregated judgments. When we look at the model outputs, however, the behavior is quite different.
> >
> > For the No-Persona baseline condition, we observe clear bimodality for the task of prediction of web-aesthetics. After the reviewer asked us to test whether this phenomenon appears in other models as well, we extended the analysis to other tasks and models under the same experimental setup. We find that the effect largely persists.
> >
> > Using the mean of ten trials for GPT-4o baseline, which refers to prompting the same model with the same input ten independent times, the prediction distributions are bimodal across all three tasks: ad CTR prediction, long-term video memorability prediction, and webpage likability prediction. This indicates that a neutral, unconditioned model is not sampling from a smooth continuum of viewpoints but instead oscillates between two dominant response modes.
> >
> >
> > | Model                   | Ad CTR Prediction | Long-Term Video Memorability | Webpage Likability Prediction |
> > | ----------------------- | ----------------- | ---------------------------- | ----------------------------- |
> > | **GPT-4o (No Persona)** | Bimodal           | Bimodal                      | Bimodal                       |
> > | **Qwen (No Persona)**   | Unimodal          | Bimodal or Multimodal        | Bimodal                       |
> >
> > For Qwen, long-term video memorability displays bimodal or multimodal structure, and webpage likability is also bimodal. The only exception is ad CTR prediction, where Qwen produces a unimodal distribution.
> >
> > Overall, while human judgments form unimodal, smooth distributions, No-Persona LLM outputs cluster into two peaks for most tasks and models. The additional experiments performed in response to the reviewer’s request confirm that this is not isolated to a single model family.

---

### Meta-Review · Area_Chair_ZUXJ · 2026-01-06

**Summary:**

The three reviewers liked the idea, the comprehensive experiments, and the good results compared to the baseline. Reviewers also pointed out missing discussions with literature, baseline choices, and some presentational issues. While there is room for improvement, the novelty of the paper outweighs the downside.

**Reviewer Scores:**

NA

---

### Decision · Program_Chairs · 2026-01-26

Accept (Poster)